# Local Identifying Causal Relations in the Presence of Latent Variables

**Zheng Li** [* 1]  **Zeyu Liu** [* 1]  **Feng Xie** [1]  **Hao Zhang** [2]  **Chunchen Liu** [3]  **Zhi Geng** [1]

## Abstract

We tackle the problem of identifying whether a variable is the cause of a specified target using observational data. State-of-the-art causal learning algorithms that handle latent variables typically rely on identifying the global causal structure, often represented as a partial ancestral graph (PAG), to infer causal relationships. Although effective, these approaches are often redundant and computationally expensive when the focus is limited to a specific causal relationship. In this work, we introduce novel local characterizations that are necessary and sufficient for various types of causal relationships between two variables, enabling us to bypass the need for global structure learning. Leveraging these local insights, we develop efficient and fully localized algorithms that accurately identify causal relationships from observational data. We theoretically demonstrate the soundness and completeness of our approach. Extensive experiments on benchmark networks and two real-world datasets further validate the effectiveness and efficiency of our method.

## 1. Introduction

Identifying causal relationships, known as causal discovery, plays a crucial role in various fields, including computer science (Jonas et al., 2017; Pearl, 2018; Schölkopf, 2022), sociology (Spirtes et al., 2000), epidemiology (Hernán & Robins, 2010), and neuroscience (Smith et al., 2011; Sanchez-Romero et al., 2019). A key challenge in causal discovery is determining whether one variable causes another (Pearl, 2009). For instance, in medical diagnosis, understanding the causal relationships between symptoms and diseases is crucial for accurate diagnoses and the devel-

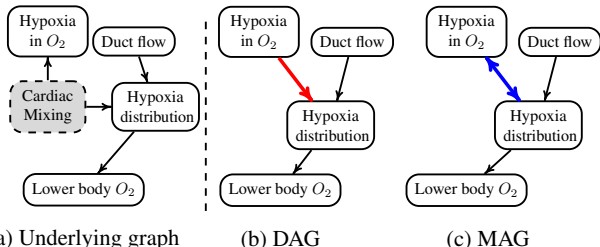

*Figure 1.* (a) The underlying graph from (Spiegelhalter et al., 1993) (CHILD network), where Cardiac Mixing is a latent variable and the other variables are observed. (b) A DAG over the observed variables in (a). (c) A MAG characterizing the causal relations over the observed variables in (a).

opment of effective treatment plans. Such insights enable healthcare professionals to efficiently identify root causes and deliver targeted interventions.

The causal graphical model is one of the most widely used models for graphically representing causal relations among observed variables, which consists of vertices denoting variables and edges denoting causal relations (Pearl, 2009; Spirtes et al., 2000). Directed acyclic graphs (DAGs) are widely used to represent causal relationships among observed variables under the assumption of causal sufficiency (Pearl, 2009; Spirtes et al., 2000), i.e. , the absence of unobserved latent variables between observed variables. However, when latent variables are present, DAGs may fail to represent the causal relationships among observed variables(Richardson & Spirtes, 2002). As shown in Figure 1 (b), the DAG incorrectly depicts Hypoxia in $O_2$ as a cause of Hypoxia distribution, but there is no directed path from Hypoxia in $O_2$ to Hypoxia distribution in the underlying graph. Therefore, directly employing methods that do not account for the influence of latent variables (when the system contains latent variables), such as those in Fang et al. (2022); Zuo et al. (2022); Zheng et al. (2024), may result in incorrect inferences about the causal relationships among the observed variables.

Maximal Ancestral Graphs (MAGs), whose main advantage is that, without explicitly including latent variables, they can represent conditional independence and causal relationships among observed variables (Richardson & Spirtes, 2002; Zhang, 2008). In a MAG, a vertex $X$ is an ancestor (cause) of a vertex $Y$ and $Y$ is a descendant (effect) of $X$ if there is a directed path from $X$ to $Y$ (Zhang, 2006). As shown in

---

*Equal contribution [1]Department of Applied Statistics, Beijing Technology and Business University, Beijing, China [2]SIAT, Chinese Academy of Sciences, Shenzhen, China [3]LingYang Co.Ltd, Alibaba Group, Hangzhou, China. Correspondence to: Feng Xie <fengxie@btbu.edu.cn>.

*Proceedings of the 42nd International Conference on Machine Learning*, Vancouver, Canada. PMLR 267, 2025. Copyright 2025 by the author(s).

Figure 1 (c), it correctly represents Hypoxia in $O_2$ does not cause Hypoxia distribution, and vice versa. However, from the observational data, without additional distributional assumptions or background knowledge, we generally learn a Markov equivalence class (MEC) of MAGs that encodes the invariant features of the underlying MAG, which can be represented by a partial ancestral graph (PAG) (Spirtes & Richardson, 1996; Zhang & Spirtes, 2005; Ali et al., 2005; Zhao et al., 2005) [1]. The undirected edges (or marks) in a PAG imply that some causal relations among variables cannot be read from the graph directly. Hence, given a Markov equivalent class of MAGs, there are three possible types of causal relationships [2]:

1. A variable $X$ is an **invariant ancestor** of a variable $Y$ if and only if there is a directed path from $X$ to $Y$ in every equivalent MAG.
2. A variable $X$ is an **invariant non-ancestor** of a variable $Y$ if and only if there is no directed path from $X$ to $Y$ in any equivalent MAG.
3. A variable $X$ is a **possible ancestor** of variable $Y$ if $X$ is neither an invariant ancestor nor an invariant non-ancestor of $Y$.

A direct approach to identifying the causal relationship between a pair of variables $(X, Y)$ is to first use methods like FCI (Spirtes et al., 2000) or RFCI (Colombo et al., 2012) to learn a PAG from observational data, and then enumerate all MAGs within this class to determine whether $X$ is an invariant ancestor or non-ancestor of $Y$ across all equivalent MAGs. However, this approach becomes computationally inefficient when the number of MAGs in the MEC is large (Malinsky & Spirtes, 2016). In addition, these approaches are often redundant and computationally expensive when the focus is limited to a specific causal relationship. In this paper, we address the challenge of locally identifying the causal relationship between a pair of variables without requiring the learning of a full PAG, the enumeration of MAGs, or the assumption of causal sufficiency. Our primary contributions are summarized as follows:

- We provide both sufficient and necessary local characterizations for the invariant ancestor, invariant non-ancestor, and possible ancestor relationships, relying solely on local structure rather than the entire graph, even in the presence of latent variables.
- We propose a novel algorithm, LocICR, that locally identifies the causal relationship between a pair of variables. We provide theoretical proof of its completeness, demonstrating that it can identify the same causal relationships

for a target pair of variables as state-of-the-art global learning approaches.
- We conduct extensive experiments on benchmark network structures and real-world datasets, showcasing the effectiveness and efficiency of our method.

## 2. Related Works

This paper focuses on identifying causal relationships between variable pairs in the presence of latent variables. Existing methods generally fall into two categories: global structure-based learning and local structure-based learning.

**Global Structure-Based Learning.** This category begins by using FCI (Spirtes et al., 2000) and its variants (Colombo et al., 2012; Claassen et al., 2013; Claassen & Heskes, 2011; Ogarrio et al., 2016; Raghu et al., 2018; Tsirlis et al., 2018; Akbari et al., 2021; Rohekar et al., 2021; Bhattacharya et al., 2021; Claassen & Bucur, 2022) to learn the global causal PAG. Then, it infers the causal relationships based on criteria proposed by Zhang (2006) and Roumpelaki et al. (2016); Mooij & Claassen (2020) or through causal effect estimation methods, such as LV-IDA (Malinsky & Spirtes, 2016) and its extensions (Maathuis et al., 2009; Nandy et al., 2017; Liu et al., 2020a; Fang & He, 2020; Pensar et al., 2020; Wang et al., 2023a). However, these methods rely on global causal graph, which can be computationally expensive and restrictive (Guo & Perkovic, 2021; Fang et al., 2022).

**Local Structure-Based Learning.** This category primarily focuses on identifying relationships between a target variable and its adjacent variables. Well-known algorithms in this line include (Yin et al., 2008; Zhou et al., 2010; Wang et al., 2014; Gao & Ji, 2015; Liu et al., 2020b; Yang et al., 2021; Liang et al., 2023). Recently, Fang et al. (2022) and its variants (Zheng et al., 2024) introduced local approaches for identifying causal relationships among arbitrary pairs of variables within a system. However, these methods require the assumption of causal sufficiency. More recently, Xie et al. (2024) proposed a method for identifying local causal structures in the presence of latent variables. Nevertheless, their approach focuses on relationships between a target variable and its adjacent variables, without generalizing to arbitrary pairs of variables.

To our knowledge, no method locally identifies the causal relationship between an arbitrary pair of variables, without assuming causal sufficiency and learning the full causal PAG.

## 3. Preliminaries

### 3.1. Terminology

A univariate variable (or vertex) is denoted by an upper-case letter (e.g., $V$), while sets of variables (or vertices) are

---

[1] Under the assumption of causal sufficiency, the Markov equivalence class (MEC) of the underlying DAG is typically represented by a completed partially directed acyclic graph (CPDAG).

[2] One may refer to Appendix A.3 for further clarification of these causal relationships.

denoted by bold uppercase letters (e.g., $\mathbf{V}$).

**Graphs.** A graph $\mathcal{G} = (\mathbf{V}, \mathbf{E})$ consists of a set of vertices $\mathbf{V} = \{V_1, \ldots, V_n\}$ and a set of edges $\mathbf{E}$. The two ends of an edge are called *marks*. A graph is **directed mixed** if the edges in the graph are **directed** ($\rightarrow$), or **bi-directed** ($\leftrightarrow$). A directed mixed graph is **ancestral** if it doesn't contain a directed or almost directed cycle. An ancestral graph is a *maximal ancestral graph* (**MAG**, denoted by $\mathcal{M}$) if for any two non-adjacent vertices, there exists a set of vertices that m-separates them. Two MAGs are **Markov equivalent** if they share the same m-separations. A class of Markov equivalent MAGs, denoted as $[\mathcal{M}]$, can be represented as a *Partially Ancestral Graph* (**PAG**, denoted by $\mathcal{P}$), where a tail '$-$' or arrowhead '$>$' occurs if the corresponding mark is tail or arrowhead in all the Markov equivalent MAGs, and a circle '$\circ$' occurs otherwise. For convenience, we use an asterisk (*) to denote any possible mark of a PAG ($\circ, >, -$) or a MAG ($>, -$). For two vertices $V_i$ and $V_j$ in $\mathcal{P}$, $V_i$ is a **possibly parent/possibly child/neighbor** of $V_j$ if there is $V_i \circ\rightarrow V_j / V_i \leftarrow\circ V_j / V_i \circ-\circ V_j$ in $\mathcal{P}$. A path is a **collider path** if every non-endpoint vertex on it is a collider along the path. A **directed path** from $V_i$ to $V_j$ is a path composed of directed edges pointing towards $V_j$. A **partially directed path** from $V_i$ to $V_j$ is a path where every edge without an arrowhead at the mark near $V_i$. A path from $V_i$ to $V_j$ that is not possibly causal is called a **non-causal path** from $V_i$ to $V_j$. The detailed graph-related definitions are provided in Appendix A.1.

**Markov Blanket.** The *Markov blanket(MB)* of a variable $X$ is the smallest set conditioned on which all other variables are statistically independent of $X$ [3] . Graphically, assuming faithfulness, in a DAG, this is the set of parents, children, and children's parents of vertex $X$. In a MAG, the Markova blanket of a vertex $X$, noted as $MB(X, \mathcal{M})$, consists of the set of parents, children, children's parents of $X$, as well as the district of $X$ and of the children of $X$, and the parents of each vertex of these districts, where the district of a vertex $V$ is the set of all vertices reachable from $V$ using only bidirected edges. Figure 2 specifically illustrates the Markov blanket of vertex $X$ in a MAG. The vertices shaded in blue belong to $MB(X, \mathcal{M})$.

Let $\mathcal{P}$ represent the MEC of $\mathcal{M}$, the MB remains invariant across $\mathcal{M}$ and $\mathcal{P}$, i.e., $MB(X, \mathcal{M}) = MB(X, \mathcal{P})$. For simplicity, $MB(X)$ denotes $MB(X, \mathcal{P})$ when unambiguous, and $MB^+(X)$ denotes $\{MB(X) \cup X\}$.

**Notations.**[4] We denote: $Pa(V_i, \mathcal{G})$, $De(V_i, \mathcal{G})$, $Adj(V_i, \mathcal{G})$ as the parent, descendant, and adjacent vertex sets of $V_i$ in $\mathcal{G}$;

---

[3]Some authors use the term "Markov blanket" without the notion of minimality, and use "Markov boundary" to denote the smallest Markov blanket. For clarity, we adopt the convention that the *Markov blanket* refers to the minimal Markov blanket.

[4]The main symbols are summarized in Table 1 in Appendix A.

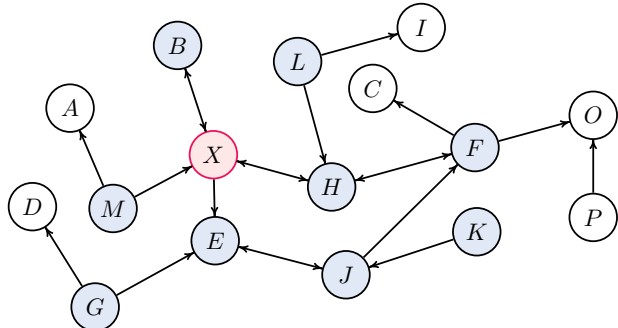

*Figure 2.* The illustrative example for $MB$, where $X$ is the target of interest and the blue vertices belong to $MB(X, \mathcal{M})$.

$Ne(V_i, \mathcal{P})$ and $PossCh(V_i, \mathcal{P})$ as the neighbor and possible child sets of $V_i$ in a PAG $\mathcal{P}$; $[\mathcal{P}]$ as the Markov equivalence class represented by PAG $\mathcal{P}$. We use the notation $\mathbf{X} \perp\!\!\!\perp \mathbf{Y}|\mathbf{Z}$ for "$\mathbf{X}$ is statistically independent of $\mathbf{Y}$ given $\mathbf{Z}$", and $\mathbf{X} \not\perp\!\!\!\perp \mathbf{Y}|\mathbf{Z}$ for the negation of the same sentence. We use the $(X, Y)$ to denote the target pair of variables, identifying whether $X$ is an invariant non-ancestor/invariant ancestor/possible ancestor of $Y$.

### 3.2. Problem Definition

Our work is in the framework of causal graphical models $\langle G, \Theta_G \rangle$, where $G$ represents the causal structure and $\Theta_G$ refers to the associated parameters with $G$ (Pearl, 2009). The causal structure $G = \langle \mathbf{V}, \mathbf{E} \rangle$ is a directed acyclic graph (DAG) where $\mathbf{V}$ represents a set of vertices and $\mathbf{E}$ represents a set of edges. The parameters $\Theta_G$ specify a functional relationship for each $V_i \in \mathbf{V}$, in the form $V_i = f_i(Pa(V_i), u_i)$, where $u_i$ represents independent errors due to omitted factors, and all error terms $u_i$ are assumed to be mutually independent. The variable set $\mathbf{V}$ consists of the observed variables $\mathbf{O}$ and the latent variables $\mathbf{L}$.

**Task.** Under the standard assumptions of the causal Markov condition and the causal Faithfulness condition, our objective is to characterize the local graphical features of different types of causal relationships between a pair of variables $X$ and $Y$, where $X, Y \in \mathbf{O}$. Subsequently, we aim to develop a fully local algorithm to determine the causal relationship between $X$ and $Y$. Note that we do not assume causal sufficiency, allowing for latent variables between observed variables in the system.

## 4. Local Characterization of Causal Relations

### 4.1. Foundations of Local Characterization

In this section, we build on the well-established local Markov property for DAGs to introduce the local Markov property for MAGs, following the ordered local Markov

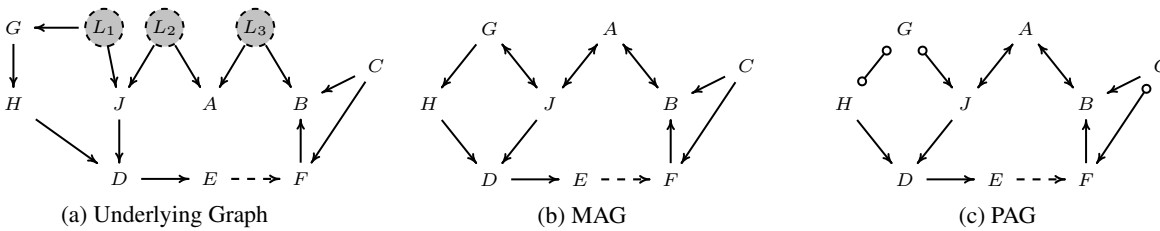

(a) Underlying Graph          (b) MAG          (c) PAG

*Figure 3.* (a) An underlying causal graph, where $L_1$, $L_2$, and $L_3$ are latent variables, dashed directed edge denotes a directed path. (b) The MAG characterizes the causal relations over the observed variables in (a). (c) The inferred PAG from observed variables.

property described in Richardson & Spirtes (2002). Specifically, the local Markov property states that a target variable is independent of its non-descendants, given *a particular set* of variables.

To define this *particular set*, we first introduce a particular type of collider path, as outlined in Definition 1.

**Definition 1** (**Arrow-Collider Path**). *In a PAG or a MAG, a path $\pi = \langle V_0, \ldots, V_n \rangle$ is called an arrow-collider path from $V_0$ to $V_n$ if every non-endpoint vertex is a collider on $\pi$, and the edge between $V_0$ and $V_1$ points into $V_0$, i.e. , $V_0 \leftrightarrow V_1 \cdots \leftarrow*V_n$. If $n = 1$, $\pi$ simplifies to $V_0 \leftarrow*V_1$.*

Building on Definition 1, we define the *particular set* graphically in Definition 2.

**Definition 2** (**Augmented Parent Set**). *Let $\mathcal{G}$ be a PAG or a MAG. The augmented parent set of a vertex $X$, denoted as $Pa^*(X, \mathcal{G})$, is defined as follows: for any vertex $V \in \mathbf{O}$, $V \in Pa^*(X, \mathcal{G})$ if and only if there exists an arrow-collider path $\pi$ from $X$ to $V$ such that:*

*(1) in a MAG, $X$ is a non-ancestor of every vertex on $\pi$, including $V$.*
*(2) in a PAG, $X$ is an invariant non-ancestor of all vertices on $\pi$, including $V$.*

In general, the augmented parent set of a target vertex includes not only its direct parents but also vertices connected via arrow-collider paths. See Example 1 for illustrations of Definitions 1 and 2.

Below, we introduce the local Markov property for MAGs using the concept of the augmented parent set. Notably, this result is theoretically equivalent to the ordered local Markov property for MAGs proposed by Richardson & Spirtes (2002). For details, see Theorem 7 in Appendix B.

**Definition 3** (**Local Markov Property for MAGs**). *Let $\mathcal{M}$ be the MAG over $\mathbf{O}$, and let $Pre(X, \mathcal{M})$ denote the pretreatment vertices of $X$ in $\mathcal{M}$, i.e. , the vertices for which $X$ is not an ancestor. The local Markov property for the MAG states that for every variable $X \in \mathbf{O}$, the following property holds:*

$$X \perp\!\!\!\perp Pre(X, \mathcal{M}) \setminus Pa^*(X, \mathcal{M}) \mid Pa^*(X, \mathcal{M}) \quad (1)$$

If no latent variables exist in the system, the local Markov property for MAGs reduces to the local Markov property

for DAGs, i.e., each variable in the DAG is independent of its non-descendants given its parents.

**Example 1.** *Consider the MAG $\mathcal{M}$ shown in Figure 3 (b), with $J$ as the target vertex of interest. Each vertex $V \in \{A, B, C, F, G\}$ is connected to $J$ by an **arrow-collider path** from $J$ to $V$. Additionally, the descendant set of $J$ in $\mathcal{M}$ is $De(J, \mathcal{M}) = \{D, E, F, B\}$. As a result, the **augmented parent set** of $J$ is $Pa^*(J, \mathcal{M}) = \{A, G\}$ and $Pre(J, \mathcal{M}) = \{A, C, G, H\}$. Therefore, the **local Markov property** is $J \perp\!\!\!\perp \{C, H\} \mid \{A, G\}$.*

### 4.2. Local Characterization

As outlined in the introduction, causal relationships between pairs of variables, inferred from non-experimental observational data can be classified into three types: invariant non-ancestors, invariant ancestors, and possible ancestors. In this section, we present local characterizations of these causal relationships, which rely on the induced subgraph of the PAG over $MB^+(X)$.

We begin by presenting the local characterization of *invariant non-ancestor relations*, as shown in Theorem 1 below.

**Theorem 1.** *Let $\mathcal{P}$ be the PAG over $\mathbf{O}$. For any pair of vertices $(X, Y)$ in $\mathcal{P}$, $X$ is an **invariant non-ancestor** of $Y$ if and only if $X \perp\!\!\!\perp Y \mid Pa^*(X, \mathcal{P})$.*

Intuitively, according to the local Markov property, given a PAG (the MEC of MAGs), $Pa^*(X, \mathcal{P})$ blocks all non-causal paths from $X$ to $Y$ in $\mathcal{P}$. If $X$ is not an invariant non-ancestor of $Y$ (i.e. , there exists a directed path from $X$ to $Y$ in a MAG $\mathcal{M} \in [\mathcal{P}]$), this implies $X \not\perp\!\!\!\perp Y \mid Pa^*(X, \mathcal{P})$. Conversely, if $X \perp\!\!\!\perp Y \mid Pa^*(X, \mathcal{P})$, then there is no directed path from $X$ to $Y$ in any $\mathcal{M} \in [\mathcal{P}]$, and $X$ is an invariant non-ancestor of $Y$.

**Example 2.** *Consider the PAG shown in Figure 3 (c), with $(J, C)$ as the target pair of interest. Each vertex in $\{A, B, C, F, G\}$ is connected to $J$ by an arrow-collider path from $J$ to it. By Definition 2, we have $Pa^*(J, \mathcal{P}) = \{A, G\}$. Furthermore, there is no active path between $J$ and $C$ given $\{A, G\}$, which implies $J \perp\!\!\!\perp C \mid \{A, G\}$. Consequently, $J$ is an invariant non-ancestor of $C$.*

We now turn to the concept of the *invariant ancestor* relation. To clarify its local characterization, we define the local

features of the *invariant ancestor* relation, categorized into two types: *explicit invariant ancestor* and *implicit invariant ancestor*, as detailed in Definitions 4 and 5, respectively.

**Definition 4** (**Explicit Invariant Ancestor**). *Given a PAG $\mathcal{P}$, a vertex $X$ is an explicit invariant ancestor of another vertex $Y$ if and only if a common directed path exists from $X$ to $Y$ in every MAG within $[\mathcal{P}]$.*

**Definition 5** (**Implicit Invariant Ancestor**). *Given a PAG $\mathcal{P}$, a vertex $X$ is an implicit invariant ancestor of another vertex $Y$ if and only if $X$ is an invariant ancestor of $Y$, but there is **no** directed path from $X$ to $Y$ common to every MAG within $[\mathcal{P}]$.*

**Remark 1.** *We define $X$ as an invariant ancestor of $Y$ if there exists a directed path from $X$ to $Y$ in every MAG in $[\mathcal{P}]$. If these directed paths are identical across all MAGs in $[\mathcal{P}]$, the invariant ancestor relation is explicit; otherwise, it is implicit. Furthermore, if a common directed path from $X$ to $Y$ exists in every MAG in $[\mathcal{P}]$, then a directed path from $X$ to $Y$ must also exist in $\mathcal{P}$ (Zhang, 2006).*

To graphically characterize these two types of invariant ancestor relations, we define a specific type of collider path in Definition 6, analogous to the *arrow-collider path*, and a *particular set* in Definition 7, similar to the *augmented parent set*.

**Definition 6** (**Circle-Collider Path**). *In a PAG, a path $\pi = \langle V_0, \ldots, V_n \rangle$ is called a circle-collider path from $V_0$ to $V_n$ if every non-endpoint vertex is a collider on $\pi$, and the edge between $V_0$ and $V_1$ is undirected relative to $V_0$, i.e. , $V_0 \circ\!\!\rightarrow V_1 \cdots \leftarrow\!\!* V_n$. If $n = 1$, $\pi$ simplifies to $V_0 \circ\!\!-\!\!* V_1$.*

**Definition 7.** *Let $\mathcal{P}$ be a PAG, the augmented undirected neighbor set of a vertex $X$, denoted as $Ne^*(X, \mathcal{P})$, is defined as follows: For any vertex $V \in \mathbf{O}$, $V \in Ne^*(X, \mathcal{P})$ if and only if there exists a circle-collider path $\pi = \langle X = V_0, V_1, \ldots, V_n = V \rangle$ from $X$ to $V$ such that for every $2 \leq i \leq n$, $X$ is an invariant non-ancestor of $V_i$* [5].

**Remark 2.** *Note that in a PAG, $Ne(X, \mathcal{P})$ consists of vertices connected to $X$ by edges of the form $\circ\!\!-\!\!\circ$, $Ne^*(X, \mathcal{P})$ includes not only $Ne(X, \mathcal{P})$ but also vertices connected by circle-collider paths.*

Next, we present the local characterization of *explicit invariant ancestor* relations, as stated in Theorem 2.

**Theorem 2.** *Let $\mathcal{P}$ be the PAG over $\mathbf{O}$. For any pair of vertices $(X, Y)$ in $\mathcal{P}$, $X$ is an **explicit invariant ancestor** of $Y$ if and only if $X \not\perp\!\!\!\perp Y \mid Pa^*(X, \mathcal{P}) \cup Ne^*(X, \mathcal{P})$.*

Intuitively, given a PAG, $Ne^*(X, \mathcal{P})$ blocks all partially directed paths(except directed paths) from $X$ to $Y$ in $\mathcal{P}$, while $Pa^*(X, \mathcal{P})$ blocks all non-causal paths that from $X$

---

to $Y$ in $\mathcal{P}$. If $X$ is not an explicit invariant ancestor of $Y$ (i.e. , no directed path from $X$ to $Y$ exists in $\mathcal{P}$), then $X \perp\!\!\!\perp Y \mid Pa^*(X, \mathcal{P}) \cup Ne^*(X, \mathcal{P})$. Conversely, if $X \not\perp\!\!\!\perp Y \mid Pa^*(X, \mathcal{P}) \cup Ne^*(X, \mathcal{P})$, a directed path from $X$ to $Y$ exists in $\mathcal{P}$, making $X$ an explicit invariant ancestor of $Y$.

**Example 3.** *Consider the PAG shown in Figure 3 (c), with $(J, B)$ be the target pair of interest. From Example 2, we know $Pa^*(J, \mathcal{P}) = \{A, G\}$. Additionally, $Ne^*(J, \mathcal{P}) = \emptyset$, as no vertex is connected to $J$ by a circle-collider path. Furthermore, there is an active path $\langle J, D, E, \ldots, F, B \rangle$ between $J$ and $B$, conditioned on $\{A, G\}$, implying $J \not\perp\!\!\!\perp B \mid \{A, G\}$. Thus, $J$ is an invariant explicit ancestor of $B$.*

We now characterize *implicit invariant ancestor* relations locally based on Definition 8, as detailed in Theorem 3.

**Definition 8.** *Let $\mathcal{P}$ be a PAG and let $\mathbb{M}$ represent the set of maximal cliques* [6] *of the induced subgraph of $\mathcal{P}$ over $PossCh(X, \mathcal{P}) \cup Ne(X, \mathcal{P})$. The set of augmented undirected neighbor of a vertex $X$ relative to a maximal clique $\mathbf{M} \in \mathbb{M}$, denoted as $Ne^*(X_\mathbf{M}, \mathcal{P})$. For any vertex $V \in \mathbf{O}$, $V \in Ne^*(X_\mathbf{M}, \mathcal{P})$ if and only if there exists a circle-collider path $\pi = \langle X = V_0, V_1, \ldots, V_n = V \rangle$ from $X$ to $V$ such that (1) for every $2 \leq i \leq n$, $X$ is an invariant non-ancestor of $V_i$ and (2) $V_1 \in \mathbf{M}$.*

**Theorem 3.** *Let $\mathcal{P}$ be the PAG over $\mathbf{O}$. For any pair of vertices $(X, Y)$ in $\mathcal{P}$, $X$ is an **implicit invariant ancestor** of $Y$ if and only if*

*(1) $X \perp\!\!\!\perp Y \mid Pa^*(X, \mathcal{P}) \cup Ne^*(X, \mathcal{P})$, but*
*(2) $X \not\perp\!\!\!\perp Y \mid Pa^*(X, \mathcal{P}) \cup Ne^*(X_\mathbf{M}, \mathcal{P})$ for every maximal clique $\mathbf{M} \in \mathbb{M}$.*

The first condition implies that there is no common directed path from $X$ to $Y$ in any MAG within $[\mathcal{P}]$. Intuitively, similar to the role of $Ne^*(X, \mathcal{P})$ in $\mathcal{P}$, $Ne^*(X_\mathbf{M}, \mathcal{P})$ can block all partially directed paths (excluding directed paths) from $X$ to $Y$ that pass through $\mathbf{M}$ in $\mathcal{P}$. If $X \not\perp\!\!\!\perp Y \mid Pa^*(X, \mathcal{P}) \cup Ne^*(X_\mathbf{M}, \mathcal{P})$ for a maximal clique $\mathbf{M} \in \mathbb{M}$, this indicates the presence of partially directed paths from $X$ to $Y$ in $\mathcal{P}$, given $Pa^*(X, \mathcal{P}) \cup Ne^*(X_\mathbf{M}, \mathcal{P})$. That is, in some MAGs within $[\mathcal{P}]$ where a directed path from $X$ to $Y$ exists, given $Pa^*(X, \mathcal{P}) \cup Ne^*(X_\mathbf{M}, \mathcal{P})$. Moreover, if $X \not\perp\!\!\!\perp Y \mid Pa^*(X, \mathcal{P}) \cup Ne^*(X_\mathbf{M}, \mathcal{P})$ holds for every $\mathbf{M} \in \mathbb{M}$, then all MAGs within $[\mathcal{P}]$ contain a directed path from $X$ to $Y$, meaning $X$ is an implicit invariant ancestor of $Y$. Conversely, if $X \perp\!\!\!\perp Y \mid Pa^*(X, \mathcal{P}) \cup Ne^*(X_\mathbf{M}, \mathcal{P})$ for a maximal clique $\mathbf{M} \in \mathbb{M}$, this implies that some MAGs within $[\mathcal{P}]$ lack a directed path from $X$ to $Y$.

**Example 4.** *Consider the PAG shown in Figure 3 (c) with $(G, D)$ as the target pair. $PossCh(G, \mathcal{P}) \cup Ne(G, \mathcal{P}) = \{H, J\}$, and the set $\mathbb{M} = \{\{H\}, \{J\}\}$. Each vertex in $\{H, J, A, B, C, F\}$ is connected to $G$ by a circle-collider*

---

[5]When $n = 1$, the condition becomes redundant and may be omitted.

[6]The definition of maximal clique is given in Appendix A.1.

*path from G to it, with G as an invariant non-ancestor of $\{A, C\}$. Thus, $Ne^*(G, \mathcal{P}) = \{H, J, A\}$. For $\mathbf{M} = \{H\}$, we have $Ne^*(G_{\mathbf{M}}, \mathcal{P}) = \{H\}$. Similarly, for $\mathbf{M} = \{J\}$, we have $Ne^*(G_{\mathbf{M}}, \mathcal{P}) = \{J, A\}$. Observing that $G \perp\!\!\!\perp D \mid \{H, J, A\}$, $G \not\perp\!\!\!\perp D \mid \{H\}$ and $G \not\perp\!\!\!\perp D \mid \{J, A\}$. Consequently, G is an invariant implicit ancestor of D.*

Based on Theorems 2 and 3, we provide a sound and complete local characterization of *invariant ancestor* relations.

**Corollary 1.** *Let $\mathcal{P}$ be the PAG over $\mathbf{O}$, and let $\mathbb{M}$ denote the set of maximal cliques of the induced subgraph of $\mathcal{P}$ over $PossCh(X, \mathcal{P}) \cup Ne(X, \mathcal{P})$. For any pair of vertices $(X, Y)$ in $\mathcal{P}$, X is an **invariant ancestor** of Y if and only if*

*(1) $X \not\perp\!\!\!\perp Y \mid Pa^*(X, \mathcal{P}) \cup Ne^*(X, \mathcal{P})$, or*
*(2) $X \not\perp\!\!\!\perp Y \mid Pa^*(X, \mathcal{P}) \cup Ne^*(X_{\mathbf{M}}, \mathcal{P})$ for every $\mathbf{M} \in \mathbb{M}$*

By Theorem 1 and Corollary 1, we can identify all stable causal relationships (invariant non-ancestor and invariant ancestor). Naturally, the remaining ones (possible ancestors) are subject to change, as stated below.

**Theorem 4.** *Let $\mathcal{P}$ be the PAG over $\mathbf{O}$. For any pair of vertices $(X, Y)$ in $\mathcal{P}$, X is an **possible ancestor** of Y if and only if neither Theorem 1 nor Corollary 1 applies.*

## 5. The Local Identifying Algorithm

This section discusses how to locally learn the Conditional Sets involved in the aforementioned theoretical results and present the algorithm for locally identifying causal relations.

### 5.1. Locally Learning Conditional Sets

We incorporate the properties of Markov Blanket (MB) to locally learn the conditional set used in the above results of local characterizations, i.e. , $Pa^*(X, \mathcal{P})$, $Ne^*(X, \mathcal{P})$, and $Ne^*(X_{\mathbf{M}}, \mathcal{P})$. Specifically, we answer the following questions:

- How to discover which vertices in $MB(X)$ are connected to $X$ via arrow-collider paths or circle-collider paths? This involves locally learning the induced subgraph of $\mathcal{P}$ over $MB^+(X)$, denoted as $\mathcal{P}_{MB^+(X)}$.
- How to determine whether $X$ is an invariant non-ancestor of the vertices on these paths? This requires identifying all vertices in $MB(X)$ for which $X$ is an invariant non-ancestor, denoted as $IPre_{MB}(X)$.

To address the first question, we extend the MMB-by-MMB algorithm proposed by Xie et al. (2024) to learn $\mathcal{P}_{MB^+(X)}$. Note that the original MMB-by-MMB algorithm focuses solely on learning the structure involving the target variable and its adjacent variables, whereas we generalize it to apply to any variable, not limited to the target variable's adjacent variables. Specially, the $\mathcal{P}_{MB^+(X)}$ learning process is iterative, with each step focusing on the local structure of a

variable $V_i$, denoted as $\mathcal{L}_{V_i}$, derived from the observed data of $MB^+(V_i)$ and is relevant for constructing $\mathcal{P}_{MB^+(X)}$. Let **Waitlist** store variables that are potentially relevant for learning $\mathcal{P}_{MB^+(X)}$, and **Donelist** store variables removed from **Waitlist** and $\mathcal{P}$ store true causal information, the basic idea is as follows:

---

Learning $\mathcal{P}_{MB^+(X)}$ algorithm

1. Given a target variable $X$, observed data $\mathbf{O}$. Initialize **Waitlist** $= \{X\}$, **Donelist** $= \emptyset$, and $\mathcal{P} = \emptyset$.
2. During each step, perform sequential iterations, focusing on the first variable $V_i$ in **Waitlist**. First, learn $MB(V_i)$, followed by the local structure $\mathcal{L}_{V_i}$ over the observed data of $MB^+(V_i)$. Next, extract true edges and direction information from $\mathcal{L}_{V_i}$ (by Proposition 1 and Proposition 2) and update $\mathcal{P}$. Orient $\mathcal{P}$ using standard orientation criteria. Then, update **Waitlist** and **Donlist**.
3. The process ends when the stopping criteria (as defined in Proposition 3) are met.

---

We outline the technical details of the algorithm below.

**Proposition 1.** *(Theorem 1 in Xie et al. (2024)) Let $X$ be any vertex in $\mathbf{O}$, and $V$ be a vertex in $MB(X)$. Then $X$ and $V$ are m-separated by a subset of $\mathbf{O} \setminus \{X, V\}$ if and only if they are m-separated by a subset of $MB(X) \setminus \{V\}$.*

Proposition 1 ensures that the existence of an edge between $X$ and another vertex $V \in MB(X)$, as identified in $\mathcal{L}_X$, is consistent with the edge identified in the PAG learned from the observed data $\mathbf{O}$.

**Proposition 2.** *Let $\mathcal{L}_X$ be the inferred PAG over $MB^+(X)$. Let $V_i$ $(1 \leq i \leq |MB(X)|)$ represent the vertices in $MB(X)$. The following statements hold:*

$\mathcal{S}1$. *The unshielded collider triples (V-structures) $V_1 * \!\!\to X \leftarrow\!\! * V_2$ identified in $\mathcal{L}_X$ are consistent with those in the ground-truth PAG.*
$\mathcal{S}2$. *The uncovered collider paths $X * \!\!\to V_1 \leftrightarrow \cdots \leftarrow\!\! * V_i$ identified in $\mathcal{L}_X$ are consistent with those in the ground-truth PAG.*

Proposition 2 specifies the locally identified colliders that are correct, representing the true directional information that can be retained.

**Proposition 3** (**Stop Rules**). *Let $X$ be the target variable of interest and **Waitlist** represent the collection of variables whose $\mathcal{L}$ will be learned. If any of the following rules are satisfied, the learned $\mathcal{P}_{MB^+(X)}$ is equivalent to the structure identified by global learning methods.*

$\mathcal{R}1$. *The edges among the variables of $MB^+(X)$ are all determined, i.e. , no circle present in the marks.*
$\mathcal{R}2$. *The **Waitlist** is empty.*

$\mathcal{R}3$. *All paths from each vertex in $MB^+(X)$, which include undirected edges (connected two vertices in $MB^+(X)$), are blocked by edges $*\!\rightarrow$.*

$\mathcal{R}1$ and $\mathcal{R}2$ indicate that all causal information of interest has been identified, or $\mathcal{L}$ for all variables in **O** has been learned. Broadly, $\mathcal{R}3$ states that if all paths connecting the undirected edges between two vertices in $MB^+(X)$ are all blocked by the edge $*\!\rightarrow$, further exploration of $\mathcal{L}$ for the remaining variables will not contribute to determining the direction of these undirected edges.

We address the second question. The basic idea is as follows:

---

Learning $IPre_{MB}(X)$ algorithm

1. Induced the subgraph $\mathcal{P}_{MB^+(X)}$ from $\mathcal{P}$. Initialize $IPre_{MB}(X)$ with vertices adjacent to $X$ where the edge is $X \leftarrow *$, and set $CandSet = MB(X) \backslash Adj(X, \mathcal{P}_{MB^+(X)})$.
2. Perform sequential iterations, focusing on the first variable $V_i$ in $CandSet$ during each step. If there exists a set $\mathbf{Z} \subseteq IPre_{MB}(X)$ such that $X \perp\!\!\!\perp V_i \mid \mathbf{Z}$, add $V_i$ to $IPre_{MB}(X)$ and remove it from $CandSet$.
3. The process ends when no variable in $CandSet$ can be added in $IPre_{MB}(X)$.

---

Proposition 4 ensures the correctness of the above algorithm.

**Proposition 4.** *Let $\mathcal{P}$ be the ground-truth PAG over* **O***, and let $\mathcal{P}_{MB^+(X)}$ denote the induced subgraph of $\mathcal{P}$ over $MB^+(X)$. Let $IPre_{MB}(X)$ be the set of invariant non-descendant of $X$. For a vertex $V \in \{MB(X) \backslash Adj(X, \mathcal{P}_{MB^+(X)})\}$, $X$ is an invariant non-ancestor of $V$ if and only if there exists a set $\mathbf{Z} \subseteq IPre_{MB}(X)$ that m-separates $X$ and $V$.*

After addressing the two main questions, we present the local learning method for conditional sets in Algorithm 1. The detailed pseudocode is provided in Algorithm 1 in Appendix D, with a complete example included in Appendix E.

**Theorem 5.** *Assuming oracle tests for conditional independence, the outputs of Algorithm 1 are identical to those obtained from the ground-truth PAG.*

### 5.2. Local Identifying Causal Relations Algorithm

In this section, we introduce the **Loc**al **I**dentifying **C**ausal **R**elations algorithm, referred to as the LocICR algorithm.

**Theorem 6.** *Assuming the oracle tests for conditional independence. Algorithm 2 is both sound and complete for identifying invariant non-ancestor, explicit invariant ancestor, implicit invariant ancestor, and possible ancestor causal relationships between any pair of variables in* **O***.*

**Complexity of Algorithm 2.** Let $r$ denote the number of local structures to be learned sequentially in Algorithm 1, and let $n$ denote the size of the observed set **O**. The worst-case complexity is $\mathcal{O}\left[\frac{r(2n-r-1)}{2} + r|MB^+|^2 2^{|MB^+|} + |\mathbb{M}|\right]$, where $|\mathbb{M}|$ is the number of maximal cliques of $PossCh(X, \mathcal{P}) \cup Ne(X, \mathcal{P})$, and $|MB^+|$ is the size of $MB^+$. The first two terms correspond to Line 1 (i.e., Algorithm 1), while the last term corresponds to Lines $2 \sim 12$. The detailed calculation process is provided in Appendix F.

---

**Algorithm 1** Local Learning Conditional Sets

**Input:** Target $X$, observed data **O**
1: The local structure $\mathcal{P}_{MB^+(X)}$ is obtained by invoking the Learning $\mathcal{P}_{MB^+(X)}$ and Learning $IPre_{MB}(X)$ algorithms.
2: Based on $\mathcal{P}_{MB^+(X)}$ and $IPre_{MB}(X)$, we derive $Pa^*(X, \mathcal{P})$, $Ne^*(X, \mathcal{P})$, and $Ne^*(X_\mathbf{M}, \mathcal{P})$ for each $\mathbf{M} \in \mathbb{M}$.

**Output:** $Pa^*(X, \mathcal{P}), Ne^*(X, \mathcal{P})$ and $Ne^*(X_\mathbf{M}, \mathcal{P})$ for each $\mathbf{M} \in \mathbb{M}$.

---

**Algorithm 2** LocICR

**Input:** Target variable pair $(X, Y)$, observed data **O**
1: $Pa^*(X, \mathcal{P})$, $Ne^*(X, \mathcal{P})$, and $Ne^*(X_\mathbf{M}, \mathcal{P})$ for each $\mathbf{M} \in \mathbb{M}$ are obtained by invoking the Algorithm 1
2: **if** $X \perp\!\!\!\perp Y \mid Pa^*(X, \mathcal{P})$ **then**
3:     **return** $X$ is an *invariant non-ancestor* of $Y$
4: **end if**
5: **if** $X \not\perp\!\!\!\perp Y \mid \{Pa^*(X, \mathcal{P}) \cup Ne^*(X, \mathcal{P})\}$ **then**
6:     **return** $X$ is an *explicit invariant ancestor* of $Y$
7: **end if**
8: $\mathbb{M}$ = the set of maximal cliques of $\{PossCh(X, \mathcal{P}) \cup Ne(X, \mathcal{P})\}$
9: **if** exist $\mathbf{M} \in \mathbb{M}$ such that $X \perp\!\!\!\perp Y \mid Pa^*(X, \mathcal{P}) \cup Ne^*(X_\mathbf{M}, \mathcal{P})$ **then**
10:     **return** $X$ is an *possible ancestor* of $Y$
11: **end if**
12: **return** $X$ is a *implicit invariant ancestor* of $Y$

**Output:** The causal relation between $X$ and $Y$.

---

## 6. Experimental Results

To evaluate the accuracy and efficiency of our algorithm, we apply it to synthetic datasets generated from benchmark networks, as well as to two real-world datasets. We utilize the existing implementation of the Total Conditioning (TC) discovery algorithm (Pellet & Elisseeff, 2008b) to identify the Markov Blanket (MB) of a variable. Our source code is available at `https://github.com/zhengli0060/LocICR`.

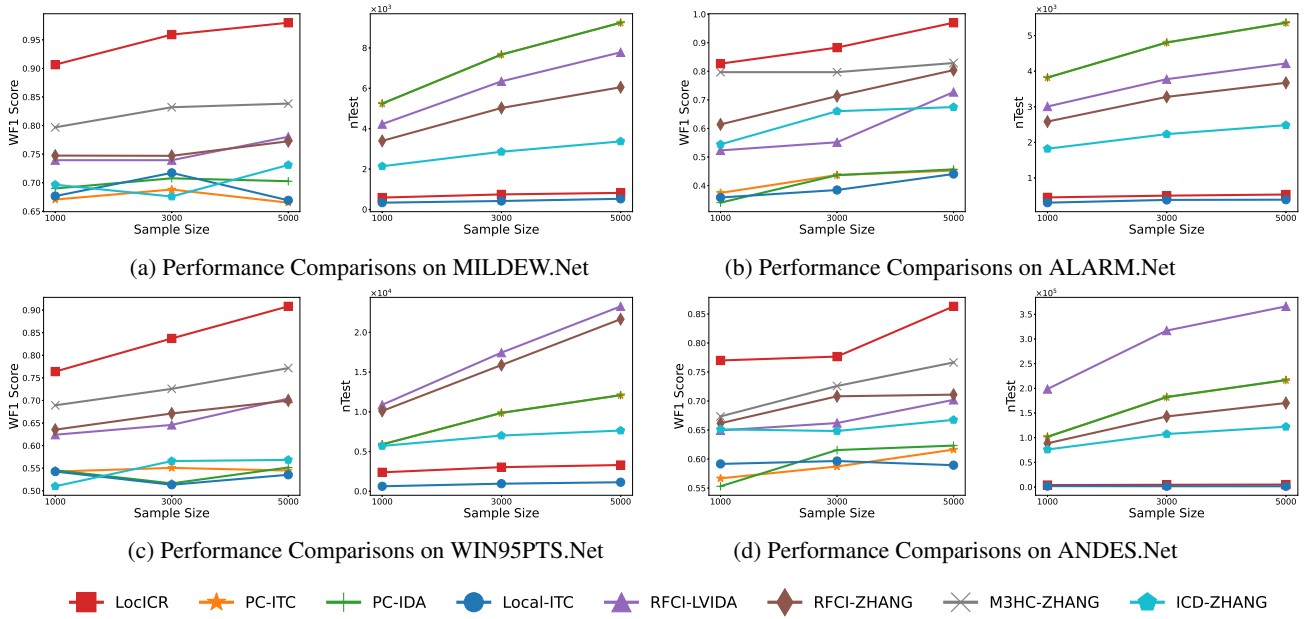

*Figure 4.* Performance of eight algorithms on four benchmark networks

## 6.1. Synthetic Data with Benchmark Networks

In this section, we compare the proposed LocICR algorithm with global structure-based learning methods, including PC-ITC (Spirtes & Glymour, 1991; Fang et al., 2022), RFCI-Zhang (Colombo et al., 2012; Zhang, 2006), M3HC-Zhang (Tsirlis et al., 2018), ICD-Zhang (Rohekar et al., 2021) (based on criteria), and PC-IDA (Maathuis et al., 2009) as well as RFCI-LVIDA (Malinsky & Spirtes, 2016)(using causal effect estimation methods). We also compared it with the local structure-based learning method, Local-ITC, which does not account for latent variables (Fang et al., 2022). Detailed descriptions of these methods are provided in Appendix G.1.

*Experimental setup:* We use four benchmark networks varying dimensionality: MILDEW, ALARM, WIN95PTS, and ANDES, containing 35, 37, 76, and 223 vertices, respectively. [7]. Following the convention in (Colombo et al., 2012; Malinsky & Spirtes, 2016; Fang et al., 2022), the benchmark networks are parameterized as linear Gaussian structural causal models, with the causal strengths chosen uniformly at random from the range $\pm(0.5, 1)$, and noises drawn from the standard Gaussian distribution. The number of latent variables is set to 4, 4, 6, and 10 for the respective network. For each network, 100 datasets were randomly generated, with latent variables randomly selected for each dataset. Two observed variables were then randomly chosen as the target pair $(X, Y)$ for each dataset. The reported

---

[7]Details of these networks can be found at https://www.bnlearn.com/bnrepository/.

results are averaged across the 100 datasets.

*Metrics:* We use the following metrics:

- **Weighted Precision (WP)**: the weighted average of per-class precision, where each precision is the ratio of true positives for the $i$-th class in the output to the total number of predictions made by the algorithm for the $i$-th class.

- **Weighted Recall (WR)**: the weighted average of per-class recall, where each recall is the ratio of true positives for the $i$-th class in the output to the total number of true instances of the $i$-th class in the ground truth.

- **Weighted F1 (WF1)**: the harmonic mean of *WP* and *WR*, calculated as
$$WF1 = \frac{2 \cdot WP \cdot WR}{WP + WR}.$$

- **nTest**: the number of (conditional) independence tests implemented by an algorithm.

*Results:* Due to space limitations, we present results for two metrics in Figure 4, with complete results provided in Appendix G. As shown, our proposed LocICR algorithm outperforms other methods in *Weighted F1* score across all networks and sample sizes, demonstrating its effectiveness. The number of conditional independence tests required by our method is significantly lower than that of global structure-based methods, such as PC-ITC, PC-IDA, RFCI-LVIDA, RFCI-ZHANG, and ICD-Zhang. Moreover, since M3HC is a hybrid method rather than a purely constraint-based one, we excluded it from the comparison regarding the nTest. Notably, although the *nTest* value of the Local-ITC method

is lower for most networks and sample sizes, our method still outperforms Local-ITC on the other three metrics. This is likely because Local-ITC only learns the local structure of $X$'s adjacencies under the assumption of no latent variables. Additionally, methods assuming causal sufficiency, such as PC-ITC, PC-IDA, and Local-ITC, show less satisfactory results, highlighting their inability to handle situations involving latent variables.

### 6.2. Application to Real-World Datasets

**General Social Survey Data.** We applied our method to a dataset from the General Social Survey, a sociological data repository available online https://gss.norc.org/us/en/gss.html. The dataset contains six observed variables: father's occupation, son's income, father's education, son's occupation, son's education, and number of siblings, with a sample size of 1380. We use the hypothesized model from Duncan et al. (1972) as a baseline. Their graph, determined using domain knowledge and temporal orders, is shown in Figure 5.

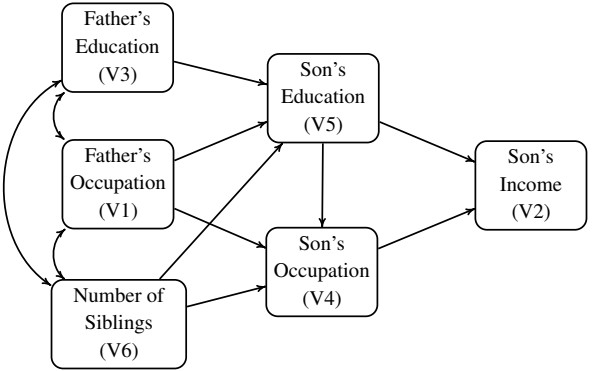

*Figure 5.* Status attainment model based on domain knowledge (Duncan et al., 1972; Shimizu et al., 2011). A directed edge between two vertices in the figure means that there could be a directed edge between the two variables. A bi-directed edge between two vertices means that the relation is not modeled.

*Results:* We selected four pairs of variables as target pairs, with direct relationships and directed paths, as well as pairs with no direct relationships or directed paths.

- We first selected the father's occupation ($X$) and the son's education ($Y$) as the target variable pair, connected by a direct edge. Our method identifies the father's occupation as an invariant ancestor of the son's education.
- Next, we selected father's occupation ($X$) and son's income ($Y$), as well as father's education ($X$) and son's income ($Y$) as the target variable pairs. In both cases, $X$ and $Y$ are connected by directed paths. Our method identifies both father's occupation and father's education as invariant ancestors of the son's income.
- Finally, for son's income ($X$) and number of siblings

($Y$), which are neither connected by a direct edge nor a directed path from $X$ to $Y$. Our method finds the son's income as an invariant non-ancestor of the number of siblings.

These findings align with the domain knowledge in Duncan et al. (1972).

**Gene Expression Data.** We applied our proposed method to the gene expression dataset from Wille et al. (2004), which contains measurements from Arabidopsis thaliana under 118 different experimental conditions, including variations in light and darkness and exposure to growth hormones. The dataset includes expression data for 33 genes. We here adopt the model presented in Wille et al. (2004) (see Figure 3 of Wille et al. (2004)) as a baseline.

*Results:* We selected six pairs of genes as target pairs, with direct relationships and directed paths, as well as pairs with no direct relationships or directed paths.

- We first selected *DXR* ($X$) and *MCT* ($Y$), as well as *HMGS* ($X$) and *HMGR1* ($Y$), as the target pairs, both of which are connected by a direct edge. Our method identifies *DXR* as an invariant ancestor of *MCT*, and similarly, *HMGS* as an invariant ancestor of *HMGR1*.
- Next, we selected *AACT1* ($X$) and *FPPS1* ($Y$), as well as *DXPS3* ($X$) and *CMK* ($Y$), where each pair is connected by a directed path from $X$ to $Y$. Our method identifies *AACT1* as an invariant ancestor of *FPPS1*, and likewise, *DXPS3* as an invariant ancestor of *CMK*.
- Finally, we considered *PPDS1* ($X$) and *DXPS1* ($Y$), as well as *DXPS1* ($X$) and *DXPS3* ($Y$), neither of which is connected by a direct edge or a directed path from $X$ to $Y$. Our method finds that *PPDS1* is an invariant non-ancestor of *DXPS1*, and similarly, *DXPS1* is an invariant non-ancestor of *DXPS3*.

These findings align with Wille et al. (2004).

## 7. Conclusion

We addressed the problem of locally learning causal relations from observational data without assuming causal sufficiency. First, we provided sufficient and necessary local characterizations for identifying invariant ancestors, invariant non-ancestors, and possible ancestors. Then, we introduced LocICR, a novel algorithm for local causal discovery. We proved that LocICR is complete, matching the accuracy of existing methods. Experiments demonstrate its effectiveness, efficiency, and robustness in handling latent variables in complex environments. Future work could explore incorporating background knowledge, such as data generation mechanisms (Kaltenpoth & Vreeken, 2023) or expert insights (Wang et al., 2023b), to enhance causal discovery within local structures in LocICR.

## Acknowledgements

We appreciate the comments from anonymous reviewers, which greatly helped to improve the paper. This research was supported by the National Natural Science Foundation of China (62306019, 62472415). Feng Xie was supported by the Beijing Key Laboratory of Applied Statistics and Digital Regulation, and the BTBU Digital Business Platform Project by BMEC.

## Impact Statement

Discovering causal relationships from observational data is fundamental for advancing scientific knowledge in settings where experimental manipulation is not feasible. Our research introduces a novel approach that focuses on locally identifying relationships of interest even in the presence of latent variables, addressing a critical limitation of existing methods that often assume no hidden confounders. A key strength of our work is the development of LocICR, an efficient and complete algorithm that utilizes necessary and sufficient local characterizations to accurately uncover causal relationships without requiring global structure learning. This makes our method both scalable and practical for complex and large-scale datasets. Our approach is particularly valuable in diverse fields such as genomics, social sciences, and network analysis, where hidden variables frequently complicate causal inference.

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

# Appendix Contents

# A. More Details on Preliminaries

Table 1. The list of main symbols used in this paper

| Symbol | Description |
|---|---|
| $G$ | A directed acyclic graph (DAG) |
| $\mathcal{M}$ | A Maximal Ancestral Graph (MAG) |
| $\mathcal{P}$ | A Partial Ancestral Graph (PAG) |
| $[\mathcal{M}]$ | A class of Markov equivalent MAGs |
| $[\mathcal{P}]$ | The Markov equivalence class represented by PAG |
| $\mathbf{V}$ | The set of all variables |
| $\mathbf{O}$ | The set of observed variables |
| $\mathbf{L}$ | The set of latent variables |
| $MB(X, G)$ | The Markov blanket of a vertex $X$ in a DAG $G$ |
| $MB(X, \mathcal{M})$ | The Markov blanket of a vertex $X$ in a MAG $\mathcal{M}$ |
| $MB(X, \mathcal{P})$, $MB(X)$ | The Markov blanket of a vertex $X$ in a PAG $\mathcal{P}$, we use $MB(X)$ to denote $MB(X, \mathcal{P})$ when there is no loss in clarity |
| $MB^+(X)$ | The set of $\{MB(X) \cup X\}$ |
| $|MB(X)|$ | Number of variables in $MB(X)$ |
| $A \quad B$ in $\mathcal{P}$ | $A$ and $B$ are not adjacent |
| $A *\!\!-\!\!* B$ in $\mathcal{P}$ | $A$ and $B$ are adjacent |
| $A \circ\!\!\to B$ in $\mathcal{P}$ | $B$ is a possible child of $A$, $A$ is a possible parent of $B$ |
| $A \circ\!\!-\!\!\circ B$ in $\mathcal{P}$ | $B$ is a neighbor of $A$, $A$ is a neighbor of $B$ |
| $A \to B$ in $\mathcal{P}$ | $A$ is a parent of $B$, $B$ is a child of $A$ |
| $A \leftrightarrow B$ in $\mathcal{P}$ | There is a latent common cause of $A$ and $B$ |
| $Pa(X, \mathcal{P})$, $Ch(X, \mathcal{P})$ | The set of all parents ($\to X$) and children ($\leftarrow X$) of $X$ in a PAG $\mathcal{P}$, respectively |
| $Adj(X, \mathcal{G})$ | The set of all variables that have direct edges connected to $X$ |
| $PossPa(X, \mathcal{P})$, $PossCh(X, \mathcal{P})$, $Ne(X, \mathcal{P})$ | The set of all possible parents ($\circ\!\!\to X$), possible children ($\leftarrow\!\!\circ X$) and neighbors ($\circ\!\!-\!\!\circ X$) of $X$ in a PAG $\mathcal{P}$, respectively |
| $Pa^*(X, \mathcal{P})$ | The augmented parent set of $X$ in a PAG $\mathcal{P}$ |
| $Ne^*(X, \mathcal{P})$ | The augmented undirected neighbor set of $X$ in a PAG $\mathcal{P}$ |
| $\mathbb{M}$ | The set of maximal cliques of the induced subgraph of $\mathcal{P}$ over $PossCh(X, \mathcal{P}) \cup Ne(X, \mathcal{P})$ |
| $\mathbf{M}$ | A maximal clique |
| $Ne^*(X_{\mathbf{M}}, \mathcal{P})$ | The augmented undirected neighbor set of $X$ relative to a maximal clique $\mathbf{M}$ |
| $\mathcal{P}_{MB^+(X)}$ | The induced subgraph of $\mathcal{P}$ over $MB^+(X)$ |
| $Pre(X, \mathcal{M})$ | The set of non-descendant variables of $X$ in a MAG $\mathcal{M}$ |
| $IPre_{MB}(X)$ | The set of vertices in $MB(X)$ such that $X$ is an invariant non-ancestor of it |
| $An(X, \mathcal{M})$, $De(X, \mathcal{M})$ | The set of all ancestors and descendants of $X$ in a MAG $\mathcal{M}$, respectively |
| $(\mathbf{X} \perp\!\!\!\perp \mathbf{Y} \mid \mathbf{Z})_{\mathcal{G}}$ | A set $\mathbf{Z}$ m-separates $\mathbf{X}$ and $\mathbf{Y}$ in $\mathcal{G}$ |
| $\mathbf{X} \perp\!\!\!\perp \mathbf{Y} \mid \mathbf{Z}$ | $\mathbf{X}$ is statistically independent of $\mathbf{Y}$ given $\mathbf{Z}$. We drop the subscript $P$ whenever it is clear from context. |
| $\mathbf{X} \not\perp\!\!\!\perp \mathbf{Y} \mid \mathbf{Z}$ | $\mathbf{X}$ is not statistically independent of $\mathbf{Y}$ given $\mathbf{Z}$ |
| $A \to B$ in $\mathcal{G}$ | $A$ is a cause of $B$, but $B$ is not a cause of $A$ |
| $A \leftrightarrow B$ in $\mathcal{G}$ | $A$ is not a cause of $B$, and $B$ is not a cause of $A$ |
| $\mathcal{L}_V$ | The local structure learned over $MB^+(V)$, utilizing the test of conditional independence and orientation of V-structures |
| $MB_{alg}$ | The algorithm used for learning $MB$ |
| $\mathbf{WaitList}$ | The list of variables to be checked by Proposition 1 and Proposition 2 |
| $\mathbf{DoneList}$ | The list of variables whose local structures have been found |

## A.1. Detailed Definitions about Graph

**Graphs.** A graph $\mathcal{G} = (\mathbf{V}, \mathbf{E})$ consists of a set of vertices $\mathbf{V} = \{V_1, \dots, V_n\}$, denoting variables, and a set of edges $\mathbf{E} \subseteq \mathbf{V} \times \mathbf{V}$, representing the relationships between variables. The two ends of an edge are called *marks*, there are three types of marks: arrowhead '<', tail '−', and circle '∘'. A graph is ***directed mixed*** if the edges in the graph are ***directed*** ($\to$), or ***bi-directed*** ($\leftrightarrow$). For two vertices $V_i$ and $V_j$ in $\mathcal{G}$, $V_i$ is a ***parent/child/spouse*** of $V_j$ if there is $V_i \to V_j / V_i \leftarrow V_j / V_i \leftrightarrow V_j$ in $\mathcal{G}$. $V_i$ and $V_j$ are adjacent if there is an edge between them. A ***path*** $p$ from $V_i$ to $V_j$ in $\mathcal{G}$ is a sequence of distinct variables $\langle V_0, \dots, V_n \rangle$ such that for $0 \leq i \leq n-1$, $V_i$ and $V_{i+1}$ are adjacent in $\mathcal{G}$. *The length of a path* equals the number of edges on

the path. A ***directed path*** from $V_i$ to $V_j$ is a path composed of directed edges pointing towards $V_j$, i.e. , $V_i \rightarrow \ldots \rightarrow V_j$. If exists a directed path from $V_i$ to $V_j$, $V_i$ is an ***ancestor*** of $V_j$ and $V_j$ is a ***descendant*** of $V_i$. An ***almost directed cycle*** happens when $V_i$ is both a spouse and an ancestor of $V_j$. A ***directed cycle*** happens when $V_i$ is both a child and an ancestor of $V_j$.

**Definition 9** (**m-connecting** (Spirtes et al., 2000; Zhang, 2008)). *In a directed mixed graph, a path $\pi$ between vertices $X$ and $Y$ is **m-connecting(acitve)** relative to a (possibly empty) set of vertices $\mathbf{Z}$ ($X, Y \notin \mathbf{Z}$) if (1) every non-collider on $\pi$ is not a member of $\mathbf{Z}$; (2) every collider on $\pi$ has a descendant in $\mathbf{Z}$.*

$\mathbf{X}$ and $\mathbf{Y}$ are said to be **m-separated** by $\mathbf{Z}$ if there is no active path between any vertex in $\mathbf{X}$ and any vertex in $\mathbf{Y}$ relative to $\mathbf{Z}$.

**Definition 10** (**Inducing Path**). *An **inducing path** between $V_i$ and $V_j$ is a path on which every non-endpoint vertex is a collider on the path and every collider is an ancestor of either $V_i$ or $V_j$.*

**Definition 11** (**Maximal Ancestral Graph** (Richardson & Spirtes, 2002; Zhang, 2008)). *A directed mixed graph is **ancestral** if it doesn't contain a directed or almost directed cycle. An ancestral graph is called **maximal** if for any two non-adjacent vertices, there is no inducing path between them, i.e. , there exists a set of vertices that m-separates them. A directed mixed graph is called a* maximal ancestral graph *(**MAG**) if it is ancestral and maximal, denoted by $\mathcal{M}$.*

A MAG is a *Directed Acyclic Graph* (DAG) if it has only directed edges. Two MAGs are ***Markov equivalent*** if they share the same m-separations.

**Definition 12** (**Partial Ancestral Graph** (Spirtes et al., 2000; Zhang, 2006)). *Let $[\mathcal{M}]$ be the Markov equivalence class of an underlying MAG $\mathcal{M}$. A partial ancestral graph (**PAG**, denoted by $\mathcal{P}$) represents the equivalence class $[\mathcal{M}]$, where a tail '−' or arrowhead '>' occurs if the corresponding mark is tail or arrowhead in every $\mathcal{M} \in [\mathcal{M}]$, and a circle '∘' occurs otherwise.*

For convenience, we use an asterisk (*) to denote any possible mark of a PAG ($\circ, >, -$) or a MAG ($>, -$). Note that $[\mathcal{P}]$ represents an equivalence class of MAGs. For two vertices $V_i$ and $V_j$ in $\mathcal{P}$, $V_i$ is a ***possibly parent/possibly child/neighbor*** of $V_j$ if there is $V_i \circ\!\!\rightarrow V_j / V_i \leftarrow\!\!\circ V_j / V_i \circ\!\!-\!\!\circ V_j$ in $\mathcal{P}$. A path is a ***collider path*** if every vertex on it (except for the endpoints) is a collider along the path. A ***partially directed path*** (possibly causal path)from $V_i$ to $V_j$ is a path where every edge without an arrowhead at the mark near $V_i$. A path from $V_i$ to $V_j$ that is not possibly causal is called a ***non-causal path*** from $V_i$ to $V_j$.

**Definition 13** (**Uncovered Path**). *A path $\pi = \langle V_0, \ldots, V_n \rangle$ is said to be **uncovered** if for every $1 \leq i \leq n - 1$, $V_{i-1}$ and $V_{i+1}$ are not adjacent, i.e. , every consecutive triple $\langle V_{i-1}, V_i, V_{i+1} \rangle$ on the path is unshielded.*

**Definition 14** (**Induced Subgraph**). *Given a graph $\mathcal{G} = \langle \mathbf{V}, \mathbf{E} \rangle$, where $\mathbf{V}$ is a vertex set and $\mathbf{E}$ is an edge set. Given a subset $\mathbf{V}'$ of $\mathbf{V}$, the* induced subgraph *of $\mathcal{G}$ over $\mathbf{V}'$ is defined as $\mathcal{G}' = \langle \mathbf{V}', \mathbf{E}' \rangle$, where $\mathbf{E}' \subset \mathbf{E}$ contains only edges between vertices in $\mathbf{V}'$.*

**Definition 15** (**Complete Graph**). *In a graph, if there is an edge between any two vertices, then the graph is called* **complete graph**.

**Definition 16** (**Maximal Clique**). *Given a graph $\mathcal{G} = (\mathbf{V}, \mathbf{E})$, a clique $\mathbf{C}$ in $\mathcal{G}$ is a subset of vertices, where $\mathbf{C} \subseteq \mathbf{V}$, such that the induced subgraph $\mathcal{G}_{\mathbf{C}}$ of $\mathcal{G}$ over $\mathbf{C}$ is a complete graph. $\mathbf{C}$ is called a **maximal clique** in $\mathcal{G}$ if there exists no clique $\mathbf{C}'$ in $\mathcal{G}$ such that $\mathbf{C}' \supset \mathbf{C}$.*

**Definition 17** (**Causal Markov Condition** (Spirtes et al., 2000)). *Given a set of variables $\mathbf{V}$ whose causal structure is represented by a DAG $G$, every variable in $\mathbf{V}$ is probabilistically independent of its non-descendants in $G$ given its parents in $G$.*

**Definition 18** (**Causal Faithfulness Condition** (Spirtes et al., 2000)). *Given a set of variables $\mathbf{V}$ whose causal structure is represented by a DAG $G$, the joint probability of $\mathbf{V}$, $P(\mathbf{V})$, is faithful to $G$ in the sense that $P(\mathbf{V})$ implies no conditional independence relations not already entailed by the causal Markov condition.*

Under the above two conditions, conditional independence relations among the observed variables correspond exactly to m-separation in the MAG or PAG $\mathcal{G}$, i.e., $(\mathbf{X} \perp\!\!\!\perp \mathbf{Y}|\mathbf{Z})_P \Leftrightarrow (\mathbf{X} \perp\!\!\!\perp \mathbf{Y}|\mathbf{Z})_{\mathcal{G}}$.

### A.2. More Details on Markov Blanket

The concept of the *Markov blanket* was first termed by Pearl (1988) and has become a widely used technique for reducing the number of variables or features, thereby enabling more efficient and robust model construction (Guyon & Elisseeff,

2003; Pellet & Elisseeff, 2008b; Gao & Ji, 2016). More specifically, the *Markov blanket* of a variable $X$ is the smallest set containing all variables carrying information about $X$ that cannot be obtained from any other variable [8] (Aliferis et al., 2010). Importantly, the *Markov blanket* possesses a unique and valuable property, which is formally described in Definition 19.

**Definition 19** (**Markov Blanket**). *The Markov blanket of a variable $X$ is the smallest set of variables $MB(X)$ such that for all $V \in \mathbf{V} \setminus (MB(X) \cup \{X\})$, $X$ is conditionally independent of $V$ given $MB(X)$:*

$$X \perp\!\!\!\perp V \mid MB(X).$$

Graphically, assuming faithfulness, the Markov blanket of a vertex $X$ in a DAG is defined as in Definition 20.

**Definition 20** (**Markov Blanket for DAGs** (Pearl, 1988; 2000))**.** *Assuming faithfulness, in a DAG, the Markov blanket of a vertex $X$ is unique, denoted as $MB(X, G)$, includes the set of parents, children, and the parents of the children (spouses) of $X$.*

The Markov blanket of one vertex in a MAG is then defined as shown in Definition 21.

**Definition 21** (**Markov Blanket for MAGs** (Richardson, 2003; Pellet & Elisseeff, 2008a))**.** *Assuming faithfulness, in a MAG, the Markov blanket of a vertex $X$, noted as $MB(X, \mathcal{M})$, consists of the set of parents, children, children's parents of $X$, as well as the district of $X$ and of the children of $X$, and the parents of each vertex of these districts, where the district of a vertex $V$ is the set of all vertices reachable from $V$ using only bidirected edges.*

### A.3. Overview of Causal Relationships

In this section, we provide an overview of the causal relationships within the Markov equivalence class of a MAG, as illustrated in Figure 6. Additionally, we present an example to clarify these causal relationships, as shown in Figure 7.

**Causal relations.** From observed data, without additional distributional assumptions, the causal relationships between a pair of variables that can generally be identified are summarized as follows, with an overview provided in Figure 6:

- **Invariant non-ancestor**: a variable X is an invariant ancestor of a variable Y if and only if *there is no directed path* from X to Y in every equivalent MAG.
- **Invariant ancestor**: a variable X is an invariant ancestor of a variable Y if and only if *there is a directed path* from X to Y in every equivalent MAG.
  - **Explicit invariant ancestor**: a variable $X$ is an explicit invariant ancestor of a variable $Y$ if and only if $X$ *has a common directed path* to $Y$ in every equivalent MAG.
  - **Implicit invariant ancestor**: a variable $X$ is an implicit invariant ancestor of a variable $Y$ if and only if $X$ is an invariant ancestor of $Y$ but *there is no common directed path* to $Y$ in every equivalent MAG.
- **Possible ancestor**: A variable X is a possible ancestor of a variable Y if X is neither an invariant ancestor nor an invariant non-ancestor of Y. In other words, $X$ is a possible ancestor of $Y$ if *there is a directed path from $X$ to $Y$ in some (but not all)* equivalent MAGs.

**Examples.** Consider the causal diagrams shown in Figure 7. We illustrate the above causal relationships with the following examples:

- **Invariant non-ancestor**: Let $X = D$ and $Y = A$. There is no directed path from $D$ to $A$ in Figure 7 (c-g), thus $D$ qualifies as an invariant non-ancestor of $A$.
- **Invariant ancestor**:
  - **Explicit invariant ancestor**: Let $X = D$ and $Y = E$. A common directed path $D \to E$ is present in every graph within Figure 7 (c-g), thereby establishing $D$ as an explicit invariant ancestor of $E$.
  - **Implicit invariant ancestor**: Let $X = A$ and $Y = D$. There is a directed path from $A$ to $D$ that can be observed in every graph within Figure 7 (c-g), but there is no common directed path from $A$ to $D$ in these graphs. Therefore, $A$ is considered an implicit invariant ancestor of $D$.
- **Possible ancestor**: Let $X = B$ and $Y = C$. A directed path from $B$ to $C$ exists in Figures 7 (f), but no directed path is found from $B$ to $C$ in Figures 7 (c-e) and (g). Consequently, $B$ is regarded as a possible ancestor of $C$.

---

[8]Some authors use the term "Markov blanket" without the notion of minimality, and use "Markov boundary" to denote the smallest Markov blanket. For clarity, we adopt the convention that the *Markov blanket* refers to the minimal Markov blanket.

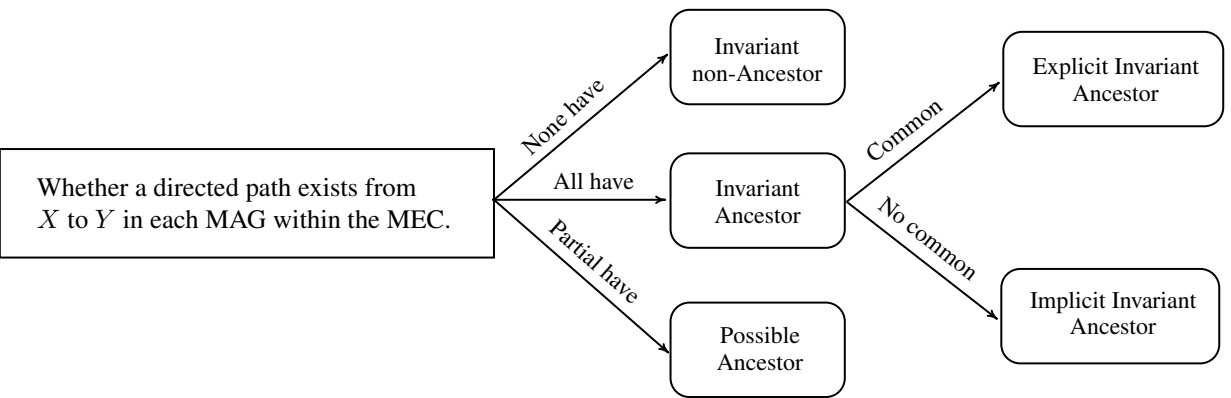

*Figure 6.* Overview of Causal Relations in the Markov equivalence class (MEC) of a true MAG.

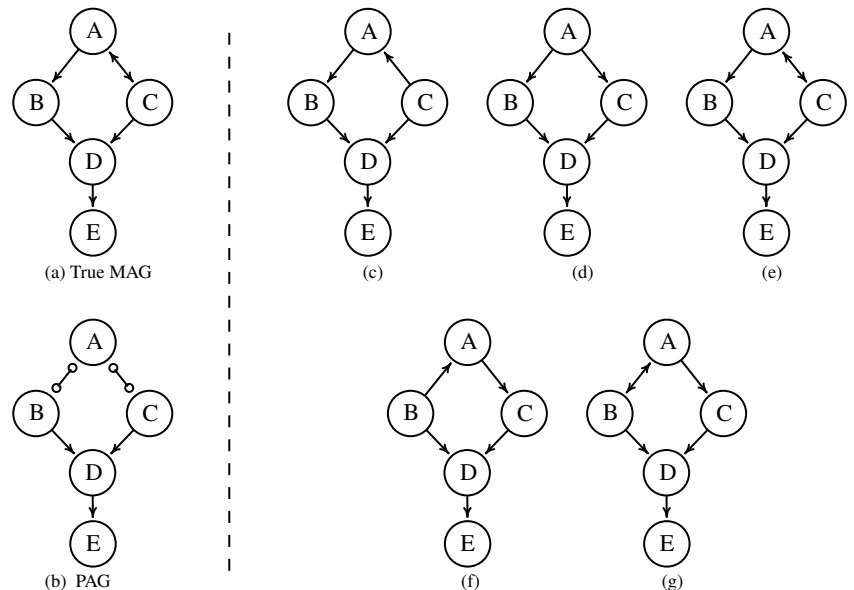

*Figure 7.* The legend illustrates the causal relationships within the equivalence class of a true MAG. (a) A true MAG. (b) The corresponding PAG of the MAG in (a). (c-g) The Markov equivalence class of MAGs represented by the PAG in (b).

## B. Supplement Theorems

In this section, we present the ordered local Markov property for MAGs as proposed by Richardson (2003). We then show Theorem 7, which states that the local Markov property in Definition 3 is consistent with the ordered local Markov property. First, we introduce the following important definitions related to the ordered local Markov property.

**Definition 22** (Total ordering). *Let $\mathcal{M}$ be the MAG over $\mathbf{O}$, the total ordering $\prec$ on the vertices of $\mathcal{M}$ is defined such that for any two vertices $V, X \in \mathbf{O}$:*

$$V \prec X \implies X \notin An(V, \mathcal{M}),$$

*where $An(V, \mathcal{M})$ denotes the set of all ancestors of $V$ in $\mathcal{M}$.*

**Definition 23** (District). *In a MAG $\mathcal{M}$, the pulsed district of $X$, denoted as $Dis^+(X, \mathcal{M})$, is the set of vertices connected to $X$ by a path consisting entirely of bidirected edges ($\leftrightarrow$), including $X$ itself. Formally:*

$$Dis^+(X, \mathcal{M}) = \{X\} \cup \{V \mid X \leftrightarrow \cdots \leftrightarrow V \text{ in } \mathcal{M}\}.$$

*The district of $X$ is defined as:*

$$Dis(X, \mathcal{M}) = Dis^+(X, \mathcal{M}) \setminus \{X\}.$$

The ordered local Markov property for MAGs proposed by Richardson (2003) is the following definition.

**Definition 24** (Ordered local Markov property)**.** *Let $\mathcal{M}$ be the MAG over $\mathbf{O}$, and let $Pre(X, \mathcal{M}) = \{V \mid V \prec X \text{ in } \mathcal{M}\}$ and $Pre^+(X, \mathcal{M}) = Pre(X, \mathcal{M}) \cup \{X\}$. Define $\mathcal{M}_{Pre^+}$ as the induced subgraph of $\mathcal{M}$ on $Pre^+(X, \mathcal{M})$. The **ordered local Markov property** states that for every variable $X \in \mathbf{O}$, the following conditional independence holds:*

$$X \perp\!\!\!\perp Pre(X, \mathcal{M}) \setminus MB(X, \mathcal{M}_{Pre^+}) \mid MB(X, \mathcal{M}_{Pre^+}) \tag{2}$$

*where $MB(X, \mathcal{M}_{Pre^+}) = Pa(Dis^+(X, \mathcal{M}_{Pre^+})) \cup Dis(X, \mathcal{M}_{Pre^+})$.*

**Lemma 1.** *Let $\mathcal{M}$ be the MAG over $\mathbf{O}$, and let $Pre^+(X, \mathcal{M}) = \{V \mid V \prec X \text{ or } V = X\}$. Let $\mathcal{M}_{Pre^+}$ denote the induced subgraph of $\mathcal{M}$ on the vertex set $Pre^+(X, \mathcal{M})$. For any vertex $X \in \mathbf{O}$, the following equivalence holds:*

$$MB(X, \mathcal{M}_{Pre^+}) \equiv Pa^*(X, \mathcal{M}).$$

*Proof.* Without loss of generality, we assume that

$$MB(X, \mathcal{M}_{Pre^+}) \neq Pa^*(X, \mathcal{M}),$$

which implies that there exists a vertex $V \in MB(X, \mathcal{M}_{Pre^+})$ such that $V \notin Pa^*(X, \mathcal{M})$, or there exists a vertex $V \notin MB(X, \mathcal{M}_{Pre^+})$ such that $V \in Pa^*(X, \mathcal{M})$.

If there exists a vertex $V \in MB(X, \mathcal{M}_{Pre^+})$, then there is one arrow-collider path from $X$ to $V$ in $\mathcal{M}$. Assume that this vertex $V \notin Pa^*(X, \mathcal{M})$, hence $X$ is an ancestor of a vertex on this path (excluding $X$) in $\mathcal{M}$. This contradicts the definition of $Pre^+(X, \mathcal{M})$. Consequently, $X$ is not an ancestor of any vertex on this path in $\mathcal{M}$, which implies $V \in Pa^*(X, \mathcal{M})$.

Conversely, if there exists a vertex $V \in Pa^*(X, \mathcal{M})$, then, according to Definition 2, we have

$$V \in Pa(Dis^+(X, \mathcal{M}_{Pre^+})) \cup Dis(X, \mathcal{M}_{Pre^+}).$$

Thus, it follows that $V \in MB(X, \mathcal{M}_{Pre^+})$.

This completes the proof. $\qquad\square$

**Theorem 7.** *Let $\mathcal{M}$ be the MAG over $\mathbf{O}$, for any vertex $X \in \mathbf{O}$, the local Markov property in Definition 3 is satisfied if and only if the ordered local Markov property is satisfied.*

*Proof.* By the definitions of the ordered local Markov property and the local Markov property, as well as Lemma 1, the equivalence is immediate. $\qquad\square$

## C. Proofs

### C.1. Proof of Theorem 1

**Lemma 2.** *(Zhang, 2006) Let $\mathcal{P}$ be the PAG over a set of variables $\mathbf{O}$ representing the invariant characteristics of equivalent MAGs, $X$ and $Y$ are distinct vertices in $\mathcal{P}$. $X$ is an invariant non-ancestor of $Y$ if and only if there is no partially directed path from $X$ to $Y$ in $\mathcal{P}$.*

*Proof.* Suppose that there is no partially directed path in $\mathcal{P}$ from $X$ to $Y$, hence there cannot be any directed path from $X$ to $Y$ in the equivalent MAGs. Therefore, $X$ is an invariant non-ancestor of $Y$. Conversely, suppose there is a partially directed path from $X$ to $Y$ in $\mathcal{P}$. This implies that there is a directed path from $X$ to $Y$ in some equivalent MAGs $\mathcal{M} \in [\mathcal{P}]$. Thus, $X$ is not an invariant non-ancestor of $Y$. $\qquad\square$

**Lemma 3** (A Crucial Property for PAG)**.** *(Lemma 3.3.1 in Zhang (2006)) In a PAG, the following property holds: for any three vertices $A$, $B$, and $C$, if $A \ast\!\!\rightarrow B \circ\!\!-\!\ast C$, then there is an edge between $A$ and $C$ with an arrowhead at $C$, namely $A \ast\!\!\rightarrow C$.*

Now, we prove the Theorem 1 based on the above Lemma 2 and Lemma 3.

*Proof.* Assume that $X \perp\!\!\!\perp Y \mid Pa^*(X, \mathcal{P})$. To begin with, assume $Y \in Pa^*(X, \mathcal{P})$. By the definition of $Pa^*(X, \mathcal{P})$, $Y$ cannot be a descendant of $X$ in any MAG $\mathcal{M} \in [\mathcal{P}]$. Hence, $X$ is an invariant non-ancestor of $Y$. Secondly, assume $Y \notin Pa^*(X, \mathcal{P})$. Since $X \perp\!\!\!\perp Y \mid Pa^*(X, \mathcal{P})$, $X$ and $Y$ are m-separated by $Pa^*(X, \mathcal{P})$. Moreover, by the definition of $Pa^*(X, \mathcal{P})$, every vertex in $Pa^*(X, \mathcal{P})$ does not block any directed path from $X$ to $Y$ for any MAG $\mathcal{M} \in [\mathcal{P}]$. Therefore, there is no partially directed path from $X$ to $Y$. If such a path existed, it would imply $X \not\perp\!\!\!\perp Y \mid Pa^*(X, \mathcal{P})$, contradicting the assumption. Thus, by Lemma 2, $X$ is an invariant non-ancestor of $Y$.

Conversely, if $X$ is an invariant non-ancestor of $Y$, then for every MAG $\mathcal{M}$ in the Markov equivalence class represented by $\mathcal{P}$, $X$ is a non-ancestor of $Y$ in every $\mathcal{M} \in [\mathcal{P}]$. If there is no active path between $X$ and $Y$, then we have $X \perp\!\!\!\perp Y \mid \emptyset$. By Definition 2, the set $Pa^*(X, \mathcal{P})$ does not open any active path between $X$ and $Y$, so $X \perp\!\!\!\perp Y \mid Pa^*(X, \mathcal{P})$ holds. If there is an active path $\pi = \langle X, V_1, \dots, V_j, Z, \dots Y \rangle$ between $X$ and $Y$, by Lemma 2, there exists an edge $V_j \leftarrow\!\!*Z$ on $\pi$. If the subpath of $\pi$ between $X$ and $V_j$ forms $X \leftarrow V_1 \cdots \leftarrow V_j$, then $\pi$ is blocked by $Pa^*(X, \mathcal{P})$. Alternatively, if the subpath of $\pi$ between $X$ and $V_j$ forms $X *\!\!-\!\!\circ V_1 \cdots *\!\!-\!\!\circ V_j$, then, by Lemma 3, there exists an edge $\leftarrow\!\!*Z$ between $Z$ and $\{X, V_1, \dots, V_j\}$, then $\pi$ also blocked by $Pa^*(X, \mathcal{P})$. Consequently, no new active path will be opened, and therefore $X \perp\!\!\!\perp Y \mid Pa^*(X, \mathcal{P})$ holds.

$\square$

## C.2. Proof of Theorem 2

Before the proof begins, we quote the following lemma since it is used to prove Theorem 2.

**Lemma 4** (Lemma 7.2 in (Maathuis & Colombo, 2015))**.** *Let $X$ and $Y$ be two distinct vertices in $\mathcal{P}$, where $\mathcal{P}$ is a PAG. If there is an uncovered partially directed path $\pi = \langle X = V_1, \dots, V_k = Y \rangle$ from $X$ to $Y$. Moreover, if $V_{i-1} *\!\!\rightarrow V_i$ for some $i \in \{2, \dots, k\}$, then $V_{j-1} \rightarrow V_j$ for all $j \in \{i+1, \dots, k\}$.*

Now, we prove the Theorem 2.

*Proof.* Assume that $X$ is an explicit invariant ancestor of $Y$. Then, there is a directed path $\pi$ from $X$ to $Y$ in $\mathcal{P}$. Moreover, this implies that there is a directed path from $X$ to $Y$ in every $\mathcal{M} \in [\mathcal{P}]$. According to Definitions 2 and 7, none of the vertices in $Pa^*(X, \mathcal{P}) \cup Ne^*(X, \mathcal{P})$ are part of the path $\pi$. Consequently, the path $\pi$ remains active conditional on $Pa^*(X, \mathcal{P}) \cup Ne^*(X, \mathcal{P})$. This implies that $X \not\perp\!\!\!\perp Y \mid Pa^*(X, \mathcal{P}) \cup Ne^*(X, \mathcal{P})$.

Suppose $X \not\perp\!\!\!\perp Y \mid Pa^*(X, \mathcal{P}) \cup Ne^*(X, \mathcal{P})$, which imply there is an active path $\pi$ between $X$ and $Y$ conditional on $Pa^*(X, \mathcal{P}) \cup Ne^*(X, \mathcal{P})$. If the length of $\pi$ is 1, then $\pi$ must be one of the following forms: $X \leftarrow\!\!*Y$, $X \circ\!\!-\!\!* Y$, and $X \rightarrow Y$. Further, if $\pi$ is one of the forms $X \leftarrow\!\!*Y$ or $X \circ\!\!-\!\!* Y$, then $Y \in Pa^*(X, \mathcal{P}) \cup Ne^*(X, \mathcal{P})$, which implies that $\pi$ is blocked by $Pa^*(X, \mathcal{P}) \cup Ne^*(X, \mathcal{P})$ and consequently, $X$ is not an explicit invariant ancestor of $Y$. If $\pi$ forms $X \rightarrow Y$, then $\pi$ is not blocked by $Pa^*(X, \mathcal{P}) \cup Ne^*(X, \mathcal{P})$ and $X$ is an explicit invariant ancestor of $Y$. If the length of $\pi$ is greater than 1, let $\pi = \langle X, V, \dots, Y \rangle$. The edge between $X$ and $V$ on $\pi$ is one of following forms: $X \leftarrow\!\!*V$, $X \circ\!\!-\!\!* V$, or $X \rightarrow V$. If the edge between $X$ and $V$ on $\pi$ is either $X \leftarrow\!\!*V$ or $X \circ\!\!-\!\!* V$, then $\pi$ is blocked by $Pa^*(X, \mathcal{P}) \cup Ne^*(X, \mathcal{P})$ as $V \in Pa^*(X, \mathcal{P}) \cup Ne^*(X, \mathcal{P})$. If the edge between $X$ and $V$ on $\pi$ is either $X \leftrightarrow V$ or $X \circ\!\!\rightarrow V$, a new active path $\pi'$ may be created if $V$ is a collider on $\pi'$. Let $\pi' = \langle X, V, W \dots, Y \rangle$, according to Definitions 2 and 7, $W \in Pa^*(X, \mathcal{P}) \cup Ne^*(X, \mathcal{P})$. Therefore, the new active path $\pi'$ is also blocked by $Pa^*(X, \mathcal{P}) \cup Ne^*(X, \mathcal{P})$. If $W$ is a collider in another path similar to $\pi'$, it will also be blocked by $Pa^*(X, \mathcal{P}) \cup Ne^*(X, \mathcal{P})$. If the edge between $X$ and $V$ on $\pi$ is either $X \leftarrow\!\!\circ V$ or $X \circ\!\!-\!\!\circ V$, according to Lemma 3, no active path will be opened. Hence, the edge between $X$ and $V$ on $\pi$ is $X \rightarrow V$, by Lemma 4, $\pi$ is a directed path and not blocked by $Pa^*(X, \mathcal{P}) \cup Ne^*(X, \mathcal{P})$. Consequently, if $X \not\perp\!\!\!\perp Y \mid Pa^*(X, \mathcal{P}) \cup Ne^*(X, \mathcal{P})$, then there is a directed path from $X$ to $Y$ in $\mathcal{P}$, $X$ is an explicit invariant ancestor of $Y$.

$\square$

## C.3. Proof of Theorem 3

**Lemma 5.** *(Zhang, 2006; Roumpelaki et al., 2016) Let $\mathcal{P}$ be the PAG over a set of variables $\mathbf{O}$ representing the invariant characteristics of equivalent MAGs, $X$ and $Y$ are distinct vertices in $\mathcal{P}$. $X$ is an implicit invariant ancestor of $Y$ if and only if there are two uncovered partially directed paths from $X$ to $Y$ in $\mathcal{P}$ such that the vertices adjacent to $X$ on the two paths respectively are distinct and are not adjacent.*

Without loss of generality, we say the chordless partially directed path as the minimal uncovered partially directed path,

where no edge joins any two nonconsecutive vertices on the path. Hence, Lemma 5 can be stated as follows: $X$ is an implicit invariant ancestor of $Y$ if and only if there are two chordless partially directed paths from $X$ to $Y$ in $\mathcal{P}$ such that the vertices adjacent to $X$ on the two paths respectively are distinct and are not adjacent.

Now, we prove the Theorem 3.

*Proof.* Assume that $X$ is an implicit invariant ancestor of $Y$. This implies that while there is no directed path from $X$ to $Y$ in $\mathcal{P}$, there exists a directed path $\pi$ from $X$ to $Y$ in every $\mathcal{M} \in [\mathcal{P}]$. By Theorem 2, we know that $X \perp\!\!\!\perp Y \mid Pa^*(X, \mathcal{P}) \cup Ne^*(X, \mathcal{P})$ as $X$ is not an explicit invariant ancestor of $Y$. Now, according to Lemma 5, we have that $|\mathbb{M}| \geq 2$, where $\mathbb{M} = \{\mathbf{M}_1, \mathbf{M}_2, \ldots, \mathbf{M}_{|\mathbb{M}|}\}$. For any $\mathbf{M}_i \in \mathbb{M}$, where $1 \leq i \leq |\mathbb{M}|$, there is a chordless partially directed path $\pi = \langle X, V_1, V_2 \ldots, V_n, Y \rangle$ from $X$ to $Y$ such that $V_1 \notin \mathbf{M}_i$. Since $\pi$ is a chordless partially directed path, and by Definitions 2 and 7, it follows that none of the vertices $\{V_2 \ldots, V_n, Y\}$ can be invariant non-descendants of $X$. Consequently, these vertices do not belong to $Pa^*(X, \mathcal{P}) \cup Ne^*(X_{\mathbf{M}_i}, \mathcal{P})$. Moreover, it is evident that $V_1 \notin Pa^*(X, \mathcal{P}) \cup Ne^*(X_{\mathbf{M}_i}, \mathcal{P})$. Which ensures that $\pi$ is active when conditioned on $Pa^*(X, \mathcal{P}) \cup Ne^*(X_{\mathbf{M}_i}, \mathcal{P})$. Thus, for each $\mathbf{M}_i \in \mathbb{M}$, we conclude that $X \not\perp\!\!\!\perp Y \mid Pa^*(X, \mathcal{P}) \cup Ne^*(X_{\mathbf{M}_i}, \mathcal{P})$.

Suppose $X \perp\!\!\!\perp Y \mid Pa^*(X, \mathcal{P}) \cup Ne^*(X, \mathcal{P})$, which implies $X$ is not an explicit invariant ancestor of $Y$ and $Y \notin Ch(X, \mathcal{P})$. Additionally, we have $X \not\perp\!\!\!\perp Y \mid Pa^*(X, \mathcal{P}) \cup Ne^*(X_{\mathbf{M}}, \mathcal{P})$ for each $\mathbf{M} \in \mathbb{M}$, which leads to the conclusion that $Y \notin Pa^*(X, \mathcal{P}) \cup Ne^*(X_{\mathbf{M}}, \mathcal{P})$. Hence, $Y \notin Adj(X, \mathcal{P})$. Now, we assume that $\mathbb{M} = \{\mathbf{M}_1, \mathbf{M}_2, \ldots, \mathbf{M}_{|\mathbb{M}|}\}$. If $|\mathbb{M}| = 1$, we have $Ne^*(X_{\mathbf{M}_1}, \mathcal{P}) = Ne^*(X, \mathcal{P})$, leading to a contradiction: both $X \perp\!\!\!\perp Y \mid Pa^*(X, \mathcal{P}) \cup Ne^*(X, \mathcal{P})$ and $X \not\perp\!\!\!\perp Y \mid Pa^*(X, \mathcal{P}) \cup Ne^*(X_{\mathbf{M}_1}, \mathcal{P})$ hold simultaneously. Therefore, $|\mathbb{M}| \geq 2$. Further, for each $\mathbf{M}_i \in \mathbb{M}$ where $1 \leq i \leq |\mathbb{M}|$, we have $X \not\perp\!\!\!\perp Y \mid Pa^*(X, \mathcal{P}) \cup Ne^*(X_{\mathbf{M}_i}, \mathcal{P})$, which implies that there exists an active path $\pi$ between $X$ and $Y$ that not blocked by $Pa^*(X, \mathcal{P}) \cup Ne^*(X_{\mathbf{M}_i}, \mathcal{P})$. The length of $\pi$ is greater than 1, let $\pi = \langle X, V, \ldots, Y \rangle$. If the edge between $X$ and $V$ on $\pi$ is $X \leftarrow\!\!* V$, then $\pi$ would be blocked by $Pa^*(X, \mathcal{P})$, implying that $X \not\perp\!\!\!\perp Y \mid Pa^*(X, \mathcal{P}) \cup Ne^*(X_{\mathbf{M}_i}, \mathcal{P})$ cannot hold. If the edge between $X$ and $V$ on $\pi$ is $X \rightarrow V$, $\pi$ would form a directed path by Lemma 4, which contradicts that $X \perp\!\!\!\perp Y \mid Pa^*(X, \mathcal{P}) \cup Ne^*(X, \mathcal{P})$. Hence, the edge between $X$ and $V$ on $\pi$ must be $X \circ\!\!-\!\!* V$. Assume that the edge between $X$ and $V$ on $\pi$ is $X \circ\!\!-\!\!\circ V$, and $\pi$ is not a partially directed path. Then there must be an edge $V_j \leftarrow\!\!* Z$ on $\pi$. By Lemma 3, this implies that there is also an edge $\leftarrow\!\!* Z$ between $Z$ and $\{X, V, \ldots, V_{j-1}\}$, which means that $Z \in Pa^*(X, \mathcal{P})$. Consequently, the assumption $X \not\perp\!\!\!\perp Y \mid Pa^*(X, \mathcal{P}) \cup Ne^*(X_{\mathbf{M}_i}, \mathcal{P})$ would not hold. Therefore, $\pi$ is an uncovered partially directed path. And $|\mathbb{M}| \geq 2$, we conclude that $X$ is an implicit invariant ancestor of $Y$.

$\square$

### C.4. Proof of Corollary 1

*Proof.* The result follows directly from Theorems 2 and 3. $\square$

### C.5. Proof of Theorem 4

*Proof.* $X$ is an invariant non-ancestor of $Y$ if and only if Theorem 1 holds, and $X$ is an invariant ancestor of $Y$ if and only if Corollary 1 holds. Thus, $X$ is a possible ancestor of $Y$ if and only if neither Theorem 1 nor Corollary 1 holds. $\square$

### C.6. Proof of Proposition 2

*Proof.* For the proof of the statement $\mathcal{S}1$ in Proposition 2, refer to Theorem 2 in Xie et al. (2024). We prove statement $\mathcal{S}2$ in Proposition 2 below. For notational convenience, let $\mathbf{S}_{X,Y}$ denote the set of vertices that m-separates $X$ and $Y$. Let $\mathcal{P}$ be the ground-truth PAG over $\mathbf{O}$.

Without loss of generality, suppose we identify an uncovered collider path in $\mathcal{L}_X$: $X *\!\!\rightarrow V_1 \leftrightarrow \cdots \leftarrow\!\!* V_i$. Here, all intermediate vertices are collider vertices, implying that every consecutive triplet forms an unshielded collider. Consider the first triplet $\langle X, V_1, V_2 \rangle$. Since this triplet is obtained over $MB^+(X)$, the following conditions hold: (1) $\forall \mathbf{S} \subseteq MB(X), X \not\perp\!\!\!\perp V_1 \mid \mathbf{S}$, (2) $\exists \mathbf{S}_{X,V_2} \subseteq MB(X), X \perp\!\!\!\perp V_2 \mid \mathbf{S}_{X,V_2}$, (3) $\forall \mathbf{S} \subseteq MB^+(X), V_1 \not\perp\!\!\!\perp V_2 \mid \mathbf{S}$, and (4) $V_1 \notin \mathbf{S}_{X,V_2}$. According to Proposition 1, condition (1) implies that the presence of a direct edge between $X$ and $V_1$ in $\mathcal{L}_X$ is consistent with $\mathcal{P}$. And condition (2) implies that the absence of a direct edge between $X$ and $V_2$ in $\mathcal{L}_X$ is also consistent with $\mathcal{P}$. Assume that the edge between $X$ and $V_1$ in $\mathcal{P}$ is $X *\!\!-V_1$. Then, combining conditions (3) and (4), we have $\forall \mathbf{S} \subseteq MB(X), X \not\perp\!\!\!\perp V_2 \mid \mathbf{S}$, which contradicts condition (2). Therefore, the edge between $X$ and $V_1$ must be $X *\!\!\rightarrow V_1$ in $\mathcal{P}$.

Next, assume the edge between $V_1$ and $V_2$ is spurious, meaning there is no direct edge between $V_1$ and $V_2$ in $\mathcal{P}$ but at least one active path exists between them. We denote a specific set of vertices as $\mathbf{Z} = Pa^*(V_1) \cup S_{X,V_2}$, note that $Pa^*(V_1) \subset MB(X)$ since $X \ast\!\!\to V_1$. We analyze the active paths as follows. First, if there exists an active path of the form $V_1 \leftarrow A_1 \ldots V_2$, then $A_1$ can block that path. If $A_1$ is a collider on a path $p1 : V_1 \cdots \ast \to A_1 \leftarrow \ast \ldots V_2$, due to the graph being ancestral and $V_1 \leftarrow A_1$, there must exist $V_1 \leftarrow A_2$ on $p1$. Thus $A_2$ also blocks $p1$, and if $A_2$ is also a collider on a path, it falls back to the case where $A_1$ is a collider. Second, if there exists an active path of the form $V_1 \leftrightarrow A_3 \to \ldots V_2$, then $A_3$ can block that path. If $A_3$ is a collider on a path $p2 : V_1 \cdots \ast \to A_3 \leftarrow \ast \ldots V_2$, due to the graph being ancestral and $A_3 \to \ldots V_2$, there must exist a $A_3 \leftarrow \ast A_4$ on $p2$. Thus $A_4$ also blocks $p2$, and if $A_4$ is also a collider on a path, it falls back to the case where $A_3$ or $A_1$ is a collider. Here, all $A_i$ belong to $Pa^*(V_1)$. Thrid, if there exists an active path of the form $V_1 \to \ldots V_2$, then $S_{X,V_2} \subseteq MB(X)$ can block that path due to $X \ast \to V_1$ and $V_1 \notin S_{X,V_2}$. Thus, $\mathbf{Z}$ blocks all active paths between $V_1$ and $V_2$, confirming that the edge $V_1 \leftarrow\!\ast V_2$ in $\mathcal{L}_X$ is consistent with $\mathcal{P}$.

Then, consider the second triplet $\langle V_1, V_2, V_3 \rangle$. The following conditions hold: (1) $\forall \mathbf{S} \subseteq MB(X), V_1 \not\perp\!\!\!\perp V_2 \mid \mathbf{S}$, (2) $\exists \mathbf{S}_{V_1,V_3} \subseteq MB(X), V_1 \perp\!\!\!\perp V_3 \mid \mathbf{S}_{V_1,V_3}$, (3) $\forall \mathbf{S} \subseteq MB^+(X), V_2 \not\perp\!\!\!\perp V_3 \mid \mathbf{S}$, and (4) $V_2 \notin \mathbf{S}_{V_1,V_3}$. As discussed above, the direct edge between $V_1$ and $V_2$ in $\mathcal{L}_X$ is consistent with $\mathcal{P}$, and the absence of a direct edge between $V_2$ and $V_3$ in $\mathcal{L}_X$ is also consistent with $\mathcal{P}$. Assuming the edge between $V_1$ and $V_2$ in $\mathcal{P}$ is $V_1 \ast\!-V_2$, conditions (3) and (4) lead to $\forall \mathbf{S} \subseteq MB(X), V_1 \not\perp\!\!\!\perp V_2 \mid \mathbf{S}$, which contradicts condition (2). Therefore, the edge between $X$ and $V_1$ must be $X \ast\!\!\to V_1$ in $\mathcal{P}$. Similarly, $Pa^*(V_2) \subset MB(X)$ since $X \ast\!\!\to V_1 \leftrightarrow V_2$. Thus, the direct edge between $V_2$ and $V_3$ in $\mathcal{L}_X$ is the same with $\mathcal{P}$.

The remaining unshielded collider triples along the path are similarly confirmed by the above analysis. Thus, the uncovered collider paths $X \ast\!\!\to V_i \leftrightarrow \cdots \leftarrow\!\ast V_j$ we identify in $\mathcal{L}_X$ must be consistent with those in the ground-truth PAG $\mathcal{P}$.

$\square$

### C.7. Proof of Proposition 3

Rule $\mathcal{R}1$ guarantees that all causal relationships among the variables in $MB^+(X)$—including both the edges and the directionalities—have been fully identified. Rule $\mathcal{R}2$ ensures that every variable in $\mathbf{O}$ has been effectively incorporated into the learning process, such that no variable remains available for further sequential learning. $\mathcal{R}1$ and $\mathcal{R}2$ are hold by construction. Therefore, our focus shifts to verifying the validity of Rule $\mathcal{R}3$. To support this, we first introduce the following lemma, which will play a crucial role in proving $\mathcal{R}3$ of Proposition 3.

**Lemma 6** (Lemma 2 in Xie et al. (2024)). *In a MAG $\mathcal{M}$ with a set of vertices $\mathbf{O}$, consider $K$ as a leaf vertex (i.e., $K$ is not an ancestor of any vertex in $\mathbf{O}$). Let $\mathcal{M}'$ be the new MAG obtained by removing $K$ from $\mathcal{M}$, and $\mathbf{O}'$ be the set of all vertices in $\mathcal{M}'$, then the following condition holds:*

$$P_{\mathcal{M}}(\mathbf{O}') = P_{\mathcal{M}'}(\mathbf{O}') \tag{3}$$

Lemma 6 establishes that the joint probability distribution of the remaining vertex set $\mathbf{O}'$ in the modified MAG $\mathcal{M}'$ is identical to that in the original MAG $\mathcal{M}$. In other words, the removal of the leaf vertex $K$ from $\mathcal{M}$ does not alter the joint probability distribution of the vertices in $\mathbf{O}'$. When considering a set of vertices that only includes the above-defined leaf vertex, the following intuition can be derived.

**Intuition:** let $\mathbf{O}$ be a set of observed varibles, $\mathcal{M}$ be the MAG graph over $\mathbf{O}$, and $\mathbf{X}' = \{\mathbf{O} \setminus \mathbf{Y}\}$, where $\mathbf{Y}$ represents the set of leaf vertices relative to $\mathbf{X}'$ in $\mathcal{M}$. According to Lemma 6, we can deduce that $P_{\mathcal{M}}(\mathbf{X}') = P_{\mathcal{M}'}(\mathbf{X}')$, where $\mathcal{M}'$ is the new MAG obtained by removing the leaf set $\mathbf{Y}$ from $\mathcal{M}$. Then, let $\mathbf{X}'' = \{\mathbf{O} \setminus \mathbf{Y} \cup \mathbf{Y}'\}$, where $\mathbf{Y}'$ denotes the set of leaf vertices relative to $\mathbf{X}''$ in $\mathcal{M}'$. Subsequently, using Lemma 6, we can infer $P_{\mathcal{M}'}(\mathbf{X}'') = P_{\mathcal{M}''}(\mathbf{X}'')$, where $\mathcal{M}''$ is the new MAG obtained by removing $\mathbf{Y}'$ from $\mathcal{M}'$. By iterating this process, we arrive at a local MAG $\mathcal{M}_L$, such that $P_{\mathcal{M}}(\mathbf{X}) = P_{\mathcal{M}'}(\mathbf{X}) = P_{\mathcal{M}''}(\mathbf{X}) = P_{\mathcal{M}_L}(\mathbf{X})$, where $\mathbf{X} = \{\mathbf{O} \setminus \mathbf{K}\}$, and $\mathbf{K}$ represents the set of leaf vertices removed during the iteration. It is worth noting that if we do not stop the process, $M_L$ will indeed become an empty graph. However, by suitably removing leaf nodes, we can derive the local subgraph $M_L$ of interest from the global MAG $M$.

Building on the lemma above, we now proceed to prove Rule $\mathcal{R}3$ of Proposition 3.

*Proof.* Assuming the sequential learning process terminates due to satisfying $\mathcal{R}3$, the result we learned is a sub-PAG of the global $\mathcal{P}$ over some observed variables. Roughly speaking, the learned result is a PAG of a $M_L$ that by suitably removing leaf nodes from the global MAG $\mathcal{M}$.

Let's provide a more detailed explanation of $\mathcal{R}3$. Assuming that we have identified a set $\mathbf{O}'$ ($MB^+(X) \subseteq \mathbf{O}'$) of variables around $MB^+(X)$ in the sequential learning process. For each vertex $T \in MB^+(X)$, and another vertex $V_1 \in MB^+(X)$. Assume that we identify there is an edge between $T$ and $V_1$, and it is an undirected edge. If we identify that all paths $\langle T, V_1, V_2, \ldots, V_n, K \rangle$ connected to this undirected edge possess the following characteristics: the edges between $T$ and $V_1$ on these path are undirected, while there is an edge form $V_n \rightarrow K$ or $V_n \circ\rightarrow K$ on these paths. When the undirected edge between any two variables in $MB^+(X)$ satisfies this condition, then $\mathcal{R}3$ is satisfied, thus we can conclude the sequential learning algorithm.

Then, We proceed to demonstrate why the sequential learning algorithm can be halted when $\mathcal{R}3$ is satisfied. In our learning process, we identify that the edges between $T$ and $V_1$ on the paths are undirected, while $V_n *\rightarrow K$. The edge between $V_n$ and $K$ on these paths from $T$ to $K$ in underlying $\mathcal{M}_L$ is either $V_n \leftrightarrow K$ or $V_n \rightarrow K$. Since we have identified $*\rightarrow K$, we can infer that these $K$ vertices belong to the leaf vertices of the underlying $\mathcal{M}_L$. Combining these $K$ vertices into a set $\mathbf{K}$, according to Lemma 6, we can deduce that $P_{\mathcal{M}'_L}(\mathbf{O}'') = P_{\mathcal{M}_L}(\mathbf{O}'')$, where $\mathbf{O}'' = \{\mathbf{O}' \setminus \mathbf{K}\}$ and $\mathcal{M}'_L$ is the new MAG obtained by removing $\mathbf{K}$ from $\mathcal{M}_L$. This implies that the joint probability distribution of the remaining vertices set $\mathbf{O}''$ in $\mathcal{M}'_L$ is equivalent to the joint probability distribution of the same vertex set $\mathbf{O}''$ in the $\mathcal{M}_L$. Then, through $P_{\mathcal{M}}(\mathbf{X}) = P_{\mathcal{M}'}(\mathbf{X}) = P_{\mathcal{M}''}(\mathbf{X}) = P_{\mathcal{M}_L}(\mathbf{X})$, we can get $P_{\mathcal{M}'_L}(\mathbf{O}'') = P_{\mathcal{M}}(\mathbf{O}'')$.

Assume that we failed to identify, based on the marginal distribution of $\mathbf{O}''$, that all paths connected to the undirected edge between any two variables in $MB^+(X)$ are blocked by $*\rightarrow$. Then, we will continue the learning process until all paths connected to the undirected edge between any two variables in $MB^+(X)$ are blocked by $*\rightarrow$ through the marginal distribution of $\mathbf{O}'$. To summarize, when $\mathcal{R}3$ is triggered, we can get $P_{\mathcal{M}_L}(\mathbf{O}') = P_{\mathcal{M}}(\mathbf{O}')$ which implies that continuing this algorithm will not contribute to orienting the undirected edges between any two variables in $MB^+(X)$. Hence, upon satisfaction of $\mathcal{R}3$, we can conclude the sequential learning $\mathcal{P}_{MB^+(X)}$. $\qquad\square$

## C.8. Proof of Proposition 4

*Proof.* By Lemma 5, for a vertex $V \in MB(X) \setminus Adj(X, \mathcal{P}_{MB^+(X)})$, $X$ is an invariant non-ancestor of $V$ if and only if there no partially directed path from $X$ to $V$. Assume that $X$ is an invariant non-ancestor of $V$. If there no active path between $X$ and $V$, then $X \perp\!\!\!\perp V \mid \emptyset$ ($\emptyset \subset IPre_{MB}(X)$). If an active path $\pi$ exists between $X$ and $V$, the length of $\pi$ is necessarily greater than 1, and $\pi$ cannot be a partially directed path. Thus, there is an edge $V_j \leftarrow\!* Z$ on $\pi$. If the subpath of $\pi$ between $X$ and $V_j$ forms $X \leftarrow V_1 \cdots \leftarrow V_j$, then $X$ and $V$ are $m$-separated by $V_1$, where $V_1 \in IPre_{MB}(X)$. If the subpath of $\pi$ between $X$ and $V_j$ forms $X *\!\!-\!\!\circ V_1 \cdots *\!\!-\!\!\circ V_j$, then by Lemma 3, there exists an edge $\leftarrow\!* Z$ between $Z$ and $\{X, V_1, \ldots, V_j\}$, where $Z \in IPre_{MB}(X)$. Conversely, assume $X \perp\!\!\!\perp V \mid \mathbf{Z}$, where $\mathbf{Z} \subseteq IPre_{MB}(X)$. If $\mathbf{Z} = \emptyset$, then there is no active path between $X$ and $V$, and consequently, no partially directed path from $X$ to $V$. If $\mathbf{Z} \neq \emptyset$, then there exists an active path $\pi$ between $X$ and $V$. If $\pi$ is a partially directed path from $X$ to $V$, then the subpath of $\pi$ between $X$ and some $Z \in \mathbf{Z}$ would also be a partially directed path. Hence, $\pi$ is not a partially directed path from $X$ to $V$. This completes the proof.

$\qquad\square$

## C.9. Proof of Theorem 5

*Proof.* We first prove the learned $\mathcal{P}_{MB^+(X)}$ is identical to the induced subgraph of the ground-truth PAG $\mathcal{P}$ over $MB^+(X)$. To establish the consistency between $\mathcal{P}_{MB^+(X)}$ and $\mathcal{P}$, it is imperative to demonstrate the consistency of all edges and orientations in the resulting graph $\mathcal{P}_{MB^+(X)}$.

Following Proposition 1, it is established that all edges in $\mathcal{P}_{MB^+(X)}$ are accurate. Subsequently, relying on Proposition 2, it can be inferred that all colliders in $\mathcal{P}_{MB^+(X)}$ are correct. Following Zhang's orientation methodology, the undirected edges in $\mathcal{P}_{MB^+(X)}$ are oriented by checking the presence of edges and directions in $\mathcal{P}_{MB^+(X)}$. Ultimately, by Proposition 3, if $\mathcal{R}1$ is triggered, there are no circles on the edges among the of $MB^+(X)$ in $\mathcal{P}_{MB^+(X)}$. If $\mathcal{R}2$ is triggered, it means that the final Donelist is equal to $\mathbf{O}$. If $\mathcal{R}3$ is triggered, it means that learning the $\mathcal{L}$ of the remaining variables does not help determine the undirected edges among the of $MB^+(X)$ in $\mathcal{P}_{MB^+(X)}$. Consequently, the resulting graph $\mathcal{P}_{MB^+(X)}$ is identical to the induced subgraph of the ground-truth PAG over $MB^+(X)$.

Given the $\mathcal{P}_{MB^+(X)}$ that is identical to the induced subgraph of the ground-truth PAG over $MB^+(X)$. For any vertex $V \in Adj(X, \mathcal{P}_{MB^+(X)})$, $V \in IPre_{MB}(X)$ if and only if the edge between $V$ and $X$ is $X \leftarrow\!* V$ in $\mathcal{P}_{MB^+(X)}$. For any vertex $V \in MB(X) \setminus Adj(X, \mathcal{P}_{MB^+(X)})$, according to the Proposition 4, we can completely identify whether it belongs to

$IPre_{MB}(X)$.

Thus, the learned $\mathcal{P}_{MB^+(X)}$ corresponds to the induced subgraph of the true PAG over $MB^+(X)$. Moreover, the learned $IPre_{MB}(X)$ includes all vertices in $MB(X)$ for which $X$ is an invariant non-ancestor. Consequently, the conditional sets obtained from Algorithm 1 match those derived from the ground-truth PAG. □

### C.10. Proof of Theorem 6

*Proof.* According to Theorem 5, the condition set obtained by Algorithm 1 is consistent with those derived from the true PAG. Furthermore, leveraging the sufficient and necessary local characterizations for invariant non-ancestors, invariant ancestors, and possible ancestors, i.e. , Theorem 1, Corollary 1, and Theorem 4, Algorithm 2 ensures a sound and complete identification of causal relationships between any pair of variables.

□

## D. More Details on Local Learning Conditional Sets Algorithm (in Section 5.1)

---

**Algorithm 1** Local Learning Conditional Sets

---

**Input:** Target $X$, observed data **O**

/*— Step one: Local learning $\mathcal{P}_{MB^+(X)}$—*/

1: Initialize : **Waitlist** := $\{X\}$, **Donelist** := $\emptyset$, $\mathcal{P} = \emptyset$.
2: **repeat**
3:     $V_i \leftarrow$ the head node of **Waitlist**.
4:     $MB(V_i) \leftarrow MB_{alg}(V_i)$ (See Algorithm 3).
5:     **if** $\exists V_j \in$ **Donelist**, $MB^+(V_i) \subseteq MB^+(V_j)$ **then**
6:         $\mathcal{L}_{V_i} \leftarrow$ the substructure of $\mathcal{L}_{V_j}$ over $MB^+(V_i)$;
7:     **else if** $MB^+(V_i) \subseteq$ **Donelist** **then**
8:         $\mathcal{L}_{V_i} \leftarrow$ the substructure of $\mathcal{P}$ over $MB^+(V_i)$;
9:     **else**
10:         Learn $\mathcal{L}_{V_i}$ over $MB^+(V_i)$.
11:     **end if**
12:     $\mathcal{P} \leftarrow$ select the edges connected to $V_i$ , the V-structures containing $V_i$, and the uncovered collider paths from $V_i$.
13:     $\mathcal{P} \leftarrow$ orient maximally the edge marks using the orientation rules of Zhang (2008).
14:     Add $V_i$ to **Donelist**, and remove $V_i$ from the **Waitlist**.
15:     Add $\{MB(V_i) \setminus (\mathbf{Waitlist} \cup \mathbf{Donelist})\}$ to **Waitlist**
16: **until** One of the stop Rules $\mathcal{R}1 \sim \mathcal{R}3$ is met

/*— Step two: Local learning $IPre_{MB}(X)$ *—/

17: $\mathcal{P}_{MB^+(X)} \leftarrow$ induced subgraph from $\mathcal{P}$
18: $IPre_{MB}(X) = \{V \in MB(X) \mid X \leftarrow *V \text{ in } \mathcal{P}_{MB^+(X)}\}$
19: $CandSet = MB(X) \setminus Adj(X, \mathcal{P}_{MB^+(X)})$.
20: **repeat**
21:     $V \leftarrow$ the head variable of $CandSet$;
22:     **if** $\exists \mathbf{Z} \subseteq IPre_{MB}(X), X \perp\!\!\!\perp V \mid \mathbf{Z}$ **then**
23:         Add $V$ to $IPre_{MB}(X)$, and remove $V$ from the $CandSet$.
24:     **end if**
25: **until** No variable in $CandSet$ can be added in $IPre_{MB}(X)$.

/*— Step three: Obtain conditional sets *—/

26: According to $\mathcal{P}_{MB^+(X)}$, $IPre_{MB}(X)$, obtain $Pa^*(X, \mathcal{P})$, $Ne^*(X, \mathcal{P})$, $Ne^*(X_\mathbf{M}, \mathcal{P})$ for each $\mathbf{M} \in \mathbb{M}$.
**Output:** $Pa^*(X, \mathcal{P})$, $Ne^*(X, \mathcal{P})$ and $Ne^*(X_\mathbf{M}, \mathcal{P})$ for each $\mathbf{M} \in \mathbb{M}$.

---

# E. Illustration of Algorithms

In this section, we illustrate our LocICR algorithm with the graph in Figure 8(a). Here we are interested in the causal relationship between the target variable pair $(J, F)$. We assume oracle tests for conditional independence.

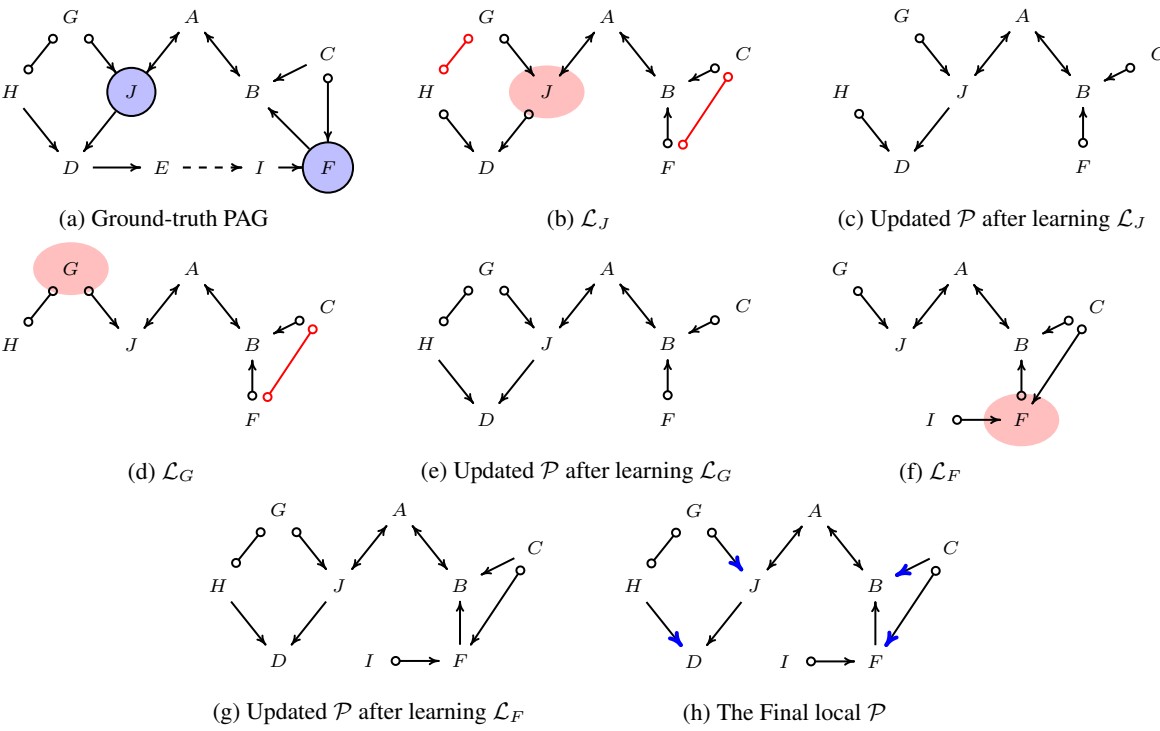

(a) Ground-truth PAG      (b) $\mathcal{L}_J$      (c) Updated $\mathcal{P}$ after learning $\mathcal{L}_J$

(d) $\mathcal{L}_G$      (e) Updated $\mathcal{P}$ after learning $\mathcal{L}_G$      (f) $\mathcal{L}_F$

(g) Updated $\mathcal{P}$ after learning $\mathcal{L}_F$      (h) The Final local $\mathcal{P}$

*Figure 8.* The sequential process of step one in Algorithm 1 in the example, where the red edges indicate that the current local results cannot be guaranteed to be consistent with the global learning results.

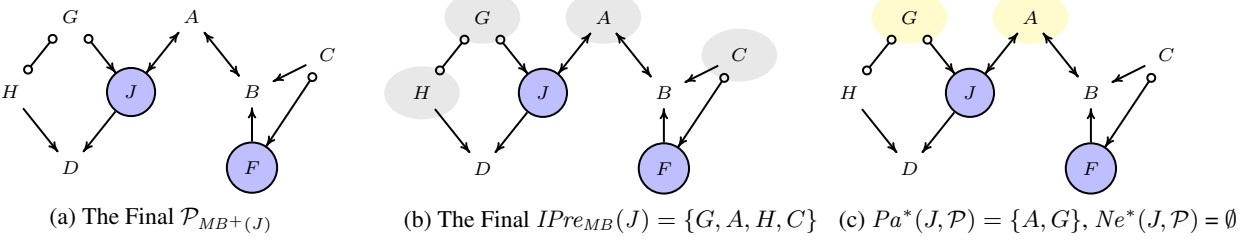

(a) The Final $\mathcal{P}_{MB+(J)}$      (b) The Final $IPre_{MB}(J) = \{G, A, H, C\}$      (c) $Pa^*(J, \mathcal{P}) = \{A, G\}$, $Ne^*(J, \mathcal{P}) = \emptyset$

*Figure 9.* The process of steps two and three in Algorithm 1 in the example.

1. **LocICR first executes the Algorithm 1, as detailed below** (Line 1 of Algorithm 2).

   **Step one: Local learning** $\mathcal{P}_{MB^+(X)}$.

   - Initialize sets: **Waitlist** $:= \{J\}$, **Donelist** $:= \emptyset$, and $\mathcal{P} = \emptyset$ (Line 1 of Algorithm 1).
   - After initialization, it runs $MB_{alg}(J)$ and obtain $MB(J) = \{A, G, F, B, C, D, H\}$ (Lines $3 \sim 4$ of Algorithm 1).
   - It then learns $\mathcal{L}_J$ over $MB^+(J)$: $H \circ\!\!-\!\!\circ G \circ\!\!\rightarrow J \leftrightarrow A \leftrightarrow B \leftarrow\!\circ C$, $H \circ\!\!\rightarrow D \leftarrow\!\circ J$, and $B \leftarrow\!\circ F \circ\!\!-\!\!\circ C$, as shown in Figure 8 (b) (Line 10 of Algorithm 1).
   - Next, it updates $\mathcal{P}$ by selecting edges connected to $J$, V-structures involving $J$, and uncovered collider paths from $J$ (Line 12 of Algorithm 1). According to Propositions 1 and 2, these edges are:
     $G \circ\!\!\rightarrow J \leftrightarrow A \leftrightarrow B \leftarrow\!\circ C$, $H \circ\!\!\rightarrow D \leftarrow\!\circ J$, and $B \leftarrow\!\circ F$.

- Orient $D \leftarrow\!\circ J$ as $D \leftarrow J$ by orientation rules (Line 13 of Algorithm 1). Consequently, it obtains $\mathcal{P}$ as shown in Figure 8 (c) .
- Update **Donelist**=$\{J\}$, and **Waitlist** $= \{A, G, F, B, C, D, H\}$ (Lines 14 $\sim$ 15 of Algorithm 1).
- Sequentially, it runs $MB_{alg}(A)$ and obtain $MB(A) = \{B, C, F, G, J\}$ . Since $MB^+(A) \subset MB^+(J)$, $\mathcal{L}_A$ is an induced substructure of $\mathcal{L}_J$ over $MB^+(A)$. (Line 6 of Algorithm 1)
- No new edges are introduced to $\mathcal{P}$ by $\mathcal{L}_A$ (Lines 12 $\sim$ 14 of Algorithm 1).
- Update **Donelist**=$\{A, J\}$, and **Waitlist** $= \{G, F, B, C, D, H\}$. (Lines 14 $\sim$ 15 of Algorithm 1)
- Run $MB_{alg}(G)$ to obtain $MB(G) = \{H, J, A, B, C, F\}$. Since $MB^+(G) \subset MB^+(J)$, $\mathcal{L}_G$ is the induced substructure of $\mathcal{L}_J$ over $MB^+(G)$, as depicted in Figure 8 (d). (Line 6 of Algorithm 1)
- Update $\mathcal{P}$ by selecting edges connected to $G$, V-structures involving $G$, and uncovered collider paths from $G$ (Line 12 of Algorithm 1). According to Propositions 1 and 2, these edges are:
  $H \circ\!\!-\!\!\circ G \circ\!\!\rightarrow J \leftrightarrow A \leftrightarrow B \leftarrow\!\circ C$, and $B \leftarrow\!\circ F$.
- Orient $D \leftarrow\!\circ H$ as $D \leftarrow H$ by orientation rules (Line 13 of Algorithm 1). Consequently, it obtains $\mathcal{P}$ as shown in Figure 8(e) .
- Update **Donelist**=$\{J, A, G\}$, and **Waitlist** $= \{F, B, C, D, H\}$.
- It then runs $MB_{alg}(F)$ and obtain $MB(F) = \{I, A, B, C, G, J\}$. (Lines 3 $\sim$ 4 of Algorithm 1)
- Learn $\mathcal{L}_F$ over $MB^+(F)$: $G \circ\!\!\rightarrow J \leftrightarrow A \leftrightarrow B \leftarrow\!\circ C$, $B \leftarrow\!\circ F \leftarrow\!\circ C$, and $F \leftarrow\!\circ I$, as depicted in Figure 8 (f) (Line 10 of Algorithm 1).
- Next, it updates $\mathcal{P}$ by selecting the edges connected to $F$, the V-structures containing $F$, and the uncovered collider paths from $F$ (Line 12 of Algorithm 1). According to Proposition 1 and Proposition 2, these edges are: $G \circ\!\!\rightarrow J \leftrightarrow A \leftrightarrow B \leftarrow\!\circ C$, $B \leftarrow\!\circ F \leftarrow\!\circ C$, and $F \leftarrow\!\circ I$.
- Orient $B \leftarrow\!\circ F$ as $B \leftarrow F$ and $C \leftarrow\!\circ F$ as $C \leftarrow F$ using orientation rules (Line 13 of Algorithm 1). Consequently, it obtains $\mathcal{P}$ as shown in Figure 8 (g).
- Then, it updates **Donelist**=$\{J, A, G, F\}$, and **Waitlist** $= \{B, C, D, H\}$.
- Finally, the repetition terminates upon reaching $\mathcal{R}3$, yielding the final PAG $\mathcal{P}$, as depicted in Figure 8 (h).

**Step two: Local learning $IPre_{MB}(X)$.**

- Obtain $\mathcal{P}_{MB^+(X)}$ induced from $\mathcal{P}$, as depicted in Figure 9 (a). (Line 17 of Algorithm 1)
- Initialize: $IPre_{MB}(J)=\{G, A\}$, $CandSet=\{H, B, C, F\}$. (Lines 18 $\sim$ 19 of Algorithm 1)
- For $H$: Since$\{G\} \subseteq IPre_{MB}(J)$, $J \perp\!\!\!\perp H \mid \{G\}$, update:
  $IPre_{MB}(J)=\{G, A, H\}$, $CandSet=\{B, C, F\}$. (Lines 22 $\sim$ 23 of Algorithm 1)
- For $C$: Since $J \perp\!\!\!\perp C \mid \emptyset$, update:
  $IPre_{MB}(J)=\{G, A, H, C\}$, $CandSet=\{B, F\}$. (Lines 22 $\sim$ 23 of Algorithm 1)
- No subset $\mathbf{Z} \subseteq IPre_{MB}(J)$ satisfies $J \perp\!\!\!\perp B \mid \mathbf{Z}$, and $J \perp\!\!\!\perp F \mid \mathbf{Z}$, thus the iteration stops. (Line 25 of Algorithm 1)
- Final result: $IPre_{MB}(J) = \{G, A, H, C\}$.

**Step three: Obtain conditional sets.** (Line 26 of Algorithm 1)

- Based on $\mathcal{P}_{MB^+(X)}$, as depicted in Figure 9 (a), each vertex in the set$\{A, B, C, F, G\}$ is connected to $J$ via an arrow-collider path from $J$ to it. Meanwhile, no vertex is linked to $J$ through a circle-collider path, implying that no vertex satisfies Definition 7.
- Among these, the vertices $\{A, G\}$ satisfy Definition 2. However, $B$ does not, as the arrow-collider path from $J$ to $B$ is $J \leftrightarrow A \leftrightarrow B$, where $B \notin IPre_{MB}(X)$. Similarly, $C$ fails to satisfy Definition 2 because the arrow-collider path connecting $J$ to $C$ is $J \leftrightarrow A \leftrightarrow B \leftarrow C$, where $B \notin IPre_{MB}(X)$. Likewise, $F$ does not satisfy Definition 2 since the arrow-collider path from $J$ to $F$ is $J \leftrightarrow A \leftrightarrow B \leftarrow F$, where $B, F \notin IPre_{MB}(X)$.
- Consequently, we derive the following results: $Pa^*(J, \mathcal{P}) = \{A, G\}$, $Ne^*(J, \mathcal{P}) = \emptyset$ and $\mathbb{M} = \emptyset$.

2. Building upon results, i.e. , $Pa^*(J, \mathcal{P}) = \{A, G\}$, $Ne^*(J, \mathcal{P}) = \emptyset$ and $\mathbb{M} = \emptyset$, LocICR identifies the causal relationship between the target variable pair $(J, F)$.

- Since $J \not\perp\!\!\!\perp F \mid \{A, G\}$, it follows that $J$ is not an invariant non-ancestor of $F$. (Line 2 of Algorithm 2)
- Since $J \not\perp\!\!\!\perp F \mid \{A, G\} \cup \emptyset$, it follows that $J$ is an explicit invariant ancestor of $F$. (Lines 5 $\sim$ 6 of Algorithm 2)

## F. Complexity of Algorithm 2

Algorithm 2's complexity mainly consists of the following two parts.

- Complexity of Algorithm 1 (Line 1 of Algorithm 2). The complexity of step one of the algorithm can be divided into two parts: the first part involves finding MB, and the second part involves learning the local structure. Let $r$ denote the number of local structures to be learned sequentially in this step. We used the TC algorithm(Pellet & Elisseeff, 2008b) to search for MB, the time complexity of finding MB among $r$ variables out of $n$ total variables is $\mathcal{O}\left(\frac{r(2n-r-1)}{2}\right)$, where $n$ denotes the size of observed set **O**. When learning local structure, we apply the logic of the PC algorithm to identify the skeleton over $MB^+(V_i)$. In the worst case, the complexity for learning a local structure over $m$ variables is $\mathcal{O}\left(m^2 2^m\right)$. Let $|MB^+|$ denote the size of $MB^+(V_i)$. In the worst case, the complexity of the step one is $\mathcal{O}\left[\frac{r(2n-r-1)}{2} + r|MB^+|^2 2^{|MB^+|}\right]$, the complexity of the step two is $\mathcal{O}\left(2^{|MB^+|}\right)$. Therefore, in the worst case, the complexity of Algorithm 1 is $\mathcal{O}\left[\frac{r(2n-r-1)}{2} + r|MB^+|^2 2^{|MB^+|}\right]$.

- Clearly, in the worst case, the complexity of Lines 2~12 of Algorithm 2 is $\mathcal{O}\left(|\mathbb{M}|\right)$, where $|\mathbb{M}|$ is the number of maximal cliques of $PossCh(X, \mathcal{P}) \cup Ne(X, \mathcal{P})$.

In conclusion, the worst-case **total complexity** is given by:

$$\mathcal{O}\left[\frac{r(2n-r-1)}{2} + r|MB^+|^2 2^{|MB^+|} + |\mathbb{M}|\right].$$

## G. More Details on Experimental Results

### G.1. Overview of Comparison Methods

The following provides a detailed description of the comparison methods:

- **PC-ITC**: This method uses the PC-stable algorithm (Colombo et al., 2014) to learn the global causal structure, followed by the global-ITC method (Fang et al., 2022) to identify the causal relationships of interest.

- **RFCI-Zhang, M3HC-Zhang, ICD-Zhang**: The RFCI (Colombo et al., 2014), M3HC (Tsirlis et al., 2018), ICD (Rohekar et al., 2021) algorithms are employed to learn the global causal structure, and the criteria outlined in (Zhang, 2006; Roumpelaki et al., 2016) are then applied to identify the causal relations of interest.

- **PC-IDA**: The PC-stable algorithm (Colombo et al., 2014) is first used to learn the global causal structure. Causal effects are then estimated using the IDA algorithm (Maathuis et al., 2009), followed by identification of the causal relationships of interest based on the causal effect testing framework in Fang et al. (2022).

- **RFCI-LVIDA**: This method utilizes the RFCI algorithm (Colombo et al., 2014) to learn the global causal structure, followed by the LV-IDA algorithm (Malinsky & Spirtes, 2016) for estimating causal effects. Finally, the causal relations of interest are identified using the causal effect testing framework from Fang et al. (2022).

- **Local-ITC**: The MB-by-MB algorithm (Wang et al., 2014) is applied to learn the local causal structure of the target variable $X$, followed by the local-ITC method (Fang et al., 2022) to identify the causal relationships of interest.

For the LV-IDA algorithm, we utilized the R implementation available at https://github.com/dmalinsk/lv-ida, along with the RFCI and PC algorithms provided in the R package *pcalg* (Kalisch et al., 2012). The ICD algorithm was implemented using the Python code from https://github.com/IntelLabs/causality-lab, while the M3HC algorithm was implemented in MATLAB using the repository at https://github.com/mensxmachina/M3HC.

### G.2. Evaluation Metrics

We evaluate the performance of the classifier using several commonly used metrics for multi-class classification problems. These include Weighted Precision (WP), Weighted Recall (WR) and Weighted F1 Score (WF1).

**Weighted Precision (WP):** The Weighted Precision is calculated as the weighted average of precision across all classes, where each class's precision is weighted by the number of true instances in that class. It is defined as:

$$\text{WP} = \frac{1}{n} \sum_{i=1}^{3} w_i \cdot \frac{TP_i}{TP_i + FP_i}$$

where:

- $TP_i$ is the number of true positives for class $i$,

- $FP_i$ is the number of false positives for class $i$,

- $w_i$ is the weight of class $i$, typically the proportion of samples belonging to that class.

**Weighted Recall (WR):** The Weighted Recall is the weighted average of recall across all classes, where each class's recall is weighted by the number of true instances in that class. It is given by:

$$\text{WR} = \frac{1}{n} \sum_{i=1}^{3} w_i \cdot \frac{TP_i}{TP_i + FN_i}$$

where:

- $FN_i$ is the number of false negatives for class $i$.

**Weighted F1 Score (WF1):** The Weighted F1 Score is the harmonic mean of Precision and Recall, weighted by the number of true instances in each class. The formula for the Weighted F1 score is:

$$\text{WF1} = \frac{1}{n} \sum_{i=1}^{3} w_i \cdot \frac{2 \cdot \text{Precision}_i \cdot \text{Recall}_i}{\text{Precision}_i + \text{Recall}_i}$$

where:

- $\text{Precision}_i$ and $\text{Recall}_i$ are the precision and recall for class $i$, respectively.

### G.3. The Markov Blanket Learning Algorithm

---

**Algorithm 3** $MB_{alg}$(Pellet & Elisseeff, 2008b)

---

**Input:** Target $V_i$, observed data of $\mathbf{O}$
1: Initialize : $MB(V_i) := \emptyset$.
2: **for** each $V_j \in \mathbf{O} \setminus V_i$ **do**
3:    **if** $V_i \not\perp V_j \mid \mathbf{O} \setminus \{V_i, V_j\}$ **then**
4:       Add $V_j$ to $MB(V_i)$
5:    **end if**
6: **end for**
7: $MB^+(V_i) = \{V_i \cup MB(V_i)\}$
**Output:** $MB^+(V_i)$

---

In this section, we describe the procedure of the Total Conditioning (TC) algorithm (Pellet & Elisseeff, 2008b), which we used to discover the Markov Blanket (MB).

**Definition 25** (Total Conditioning (Pellet & Elisseeff, 2008b)). *In the context of a faithful causal graph $\mathcal{G}$, the following holds:*

$$\forall X, Y \in \mathbf{V} : \quad (X \in \textit{Markov blanket}(Y)) \Leftrightarrow (X \not\perp Y \mid \mathbf{V} \setminus X, Y) \tag{4}$$

### G.4. Complete Results

All experiments were performed with an Intel 2.70GHz CPU and 64 GB of memory. The complete experimental results for the four benchmark Bayesian networks are presented in Figure 10. Table 2 provides a detailed overview of the network statistics used in this paper [9].

*Table 2.* Statistics on the Networks.

| Networks | Num.nodes | Number of arcs | Max in-degree | Avg degree |
|----------|-----------|----------------|---------------|------------|
| MILDEW | 35 | 46 | 3 | 2.63 |
| ALARM | 37 | 46 | 4 | 2.49 |
| WIN95PTS | 76 | 112 | 7 | 2.95 |
| ANDES | 223 | 338 | 6 | 3.03 |

---

[9]Details of these networks can be found at `https://www.bnlearn.com/bnrepository/`.

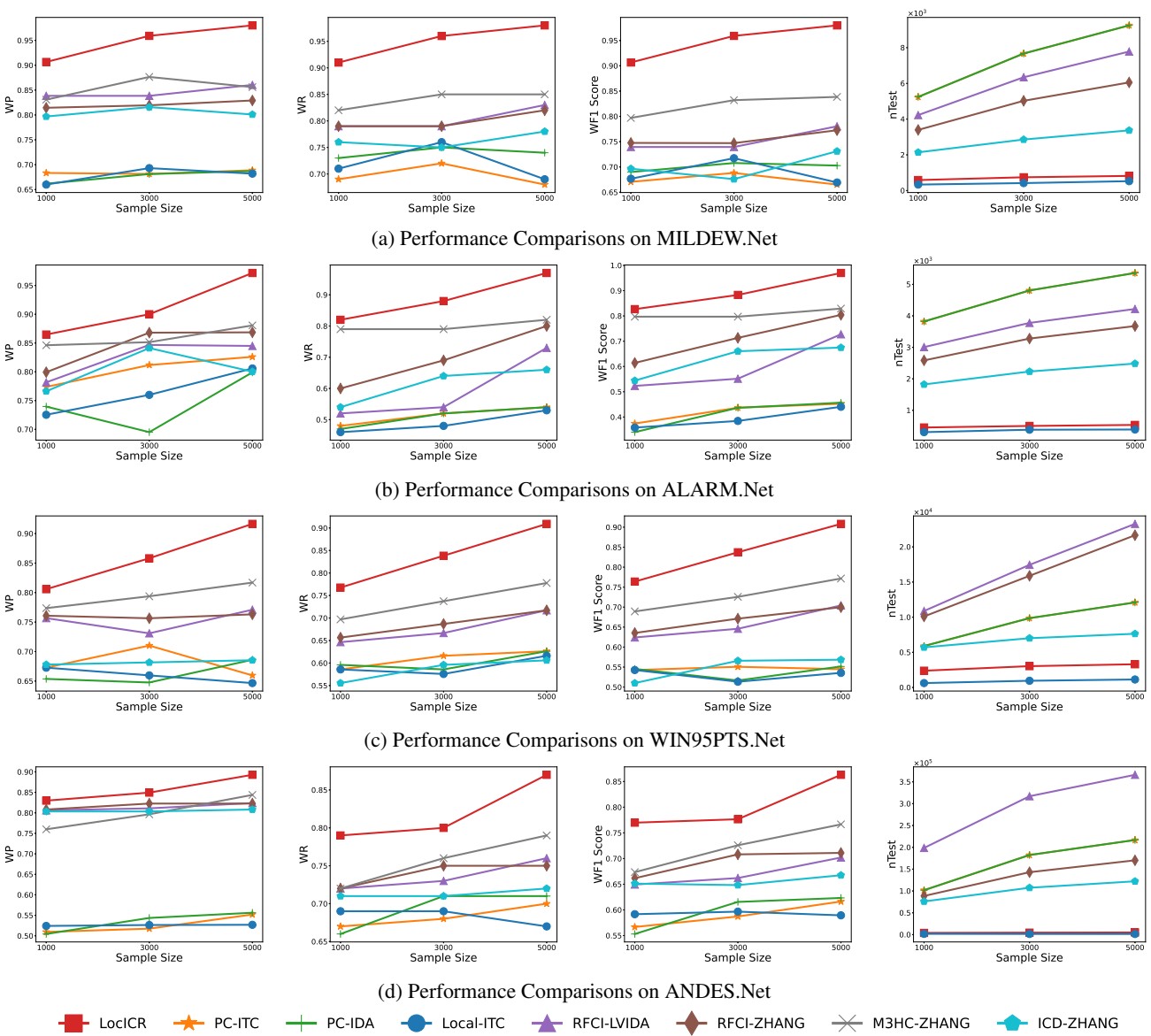

*Figure 10.* Performance of various algorithms on four benchmark networks

