# OpenReview forum: "Local Identifying Causal Relations in the Presence of Latent Variables"
_ICML.cc/2025/Conference — ICML 2025 spotlightposter_

### Official Review · Reviewer_FPtp · 2025-02-27

**Overall Recommendation:** 4

**Summary:**

The paper proposes both sufficient and necessary local characterizations for the invariant ancestor, invariant non-ancestor, and possible ancestor relationships, relying solely on local structure rather than the entire graph, even in the presence of latent variables. A novel algorithm, LocICR, leverages these local characterizations to efficiently identify causal relationships. The algorithm is proven to be sound and complete. Extensive experiments on benchmark networks and two real-world datasets demonstrate the effectiveness and efficiency of LocICR compared to existing methods.

## update after rebuttal
I've decided to raise my score from 3 to 4. The authors have provided clear and detailed responses to my concerns. For Q1, they clarified the learning process of the Markov blanket and incorporated this into the revised manuscript. For Q2, they explained the technical differences between their method and existing approaches like IDA and LV-IDA, emphasizing the advantages of their approach. For Q3, they elaborated on the distinction between the MMB-by-MMB algorithm and their Algorithm 1, highlighting the extension for identifying local characterizations. For Q4, they clarified why certain edges were not preserved, emphasizing the validation process using propositions. These thorough and thoughtful responses demonstrate a strong commitment to addressing my feedback and improving the quality of the manuscript, leading me to adjust my score upward.

**Claims And Evidence:**

Yes, the claims made in the paper are well-supported by both theoretical proofs and empirical evaluations. The authors provide solid theoretical foundations for their local characterizations (Theorems 1, 2, 3, and 4) and establish the soundness and completeness of the LocICR algorithm (Theorem 6). A comparison with several state-of-the-art methods demonstrates that LocICR outperforms these techniques in terms of accuracy and computational efficiency. Furthermore, extensive experiments on two real-world datasets further validate the practical applicability and effectiveness of the proposed approach.

**Essential References Not Discussed:**

No, the related works are thoroughly summarized and appropriately referenced.

**Experimental Designs Or Analyses:**

Yes, I have reviewed the soundness of the experimental designs and analyses. Extensive experiments on both synthetic and real-world datasets demonstrate that LocICR outperforms existing methods in terms of accuracy and computational efficiency.

**Methods And Evaluation Criteria:**

Yes, the proposed local characterizations of causal relationships and the LocICR algorithm are well-suited to address the problem at hand. The proposed method focuses on the local structures to identify causal relationships, which is a practical approach for large-scale datasets with latent variables.

**Other Comments Or Suggestions:**

- Additional details regarding the learning of $\mathcal{L}_{V_i}$ should be provided.
- Detailed technical comparisons between the proposed and compared methods (e.g., IDA, LV-IDA) are not explicitly stated in the manuscript, which should be clarified.

**Other Strengths And Weaknesses:**

**Strengths And Weaknesses**:
- The paper provides a novel and efficient approach to causal discovery, addressing significant limitations of existing methods.
- The theoretical contributions are solid, and the proposed algorithm is practical.
- The paper is well-written, with clear explanations and examples.
- Extensive experiments on both synthetic and real-world datasets show that the LocICR algorithm outperforms baselines significantly.
- The appendix provides helpful clarifications.

**Questions For Authors:**

- More details about what is the specific difference between the MMB-by-MMB algorithm and Algorithm 1.
- The edges between G and H, C and F in Fig. 7 (b) are correct, why aren't they preserved in $\mathcal{P}$?

**Relation To Broader Scientific Literature:**

The key contributions of the paper are clear, and I think represent valuable research for the causal discovery community and are novel relative to the broader scientific literature on causal discovery. Prior work such as (Fang et al., 2022), and (Xie et al., 2024) also focuses on locally identifying causal relationships, however, the method proposed by (Fang et al., 2022) requires the assumption of causal sufficiency, and the method proposed by (Xie et al., 2024) focuses on relationships between a target variable and its adjacent variables, without generalizing to arbitrary pairs of variables. The proposed LocICR algorithm addresses these limitations.

**Theoretical Claims:**

Yes, I have checked the correctness of the proofs for the main theoretical claims made in the paper.

---

> ### Author Rebuttal · Authors · 2025-04-01
>
> We are deeply grateful for the time you devoted to reviewing our manuscript. We appreciate that you think our proposal is a very solid paper. We hope that the following responses adequately address your concerns.
>
> **Q1.** ``Additional details regarding the learning of $\mathcal{L}_{V_i}$ should be provided.''
>
> **A1.** Thanks for being thoughtful. Specifically, the learning process of $\mathcal{L}_{V_i}$ first learns the Markov blanket of $V_i$, then performs skeleton learning based on the variables in $MB^{+}(V_i)$, and finally determines the V-structures. As you suggested, we have incorporated this clarification in the revised version.
>
> **Q2.** ``Detailed technical comparisons between the proposed and compared methods (e.g., IDA, LV-IDA) are not explicitly stated in the manuscript, which should be clarified.''
>
> **A2.** Thanks for the helpful suggestion. After a PAG (or CPDAG) is learned, LV-IDA（or IDA）enumerates all corresponding MAGs (or DAGs), identifies adjustment sets that satisfy the generalized back-door criterion, and estimates causal effects. If all these estimated effects are consistently nonzero (or consistently zero), an invariant ancestor (or non-ancestor) is determined. By contrast, our method does not require global graph learning or enumeration. We will include the above discussion.
>
> **Q3.** ``More details about what is the specific difference between the MMB-by-MMB algorithm and Algorithm 1.''
>
> **A3.** Thank you for raising this point. The MMB-by-MMB algorithm focuses solely on learning the local structure involving the target variable and its adjacent variables. To identify local characterizations (Theorems 1, 2, 3, 4), we extend the MMB-by-MMB algorithm to learn the $\mathcal{P}_{MB^+(X)}$. As a result, we can identify causal relationships between any pair of variables, no longer limited to the target variable and its adjacent variables.
>
>
> **Q4** ``The edges between G and H, C and F in Fig. 7 (b) are correct, why aren't they preserved in $\mathcal{P}$  ?''
>
> **A4.** Thank you for your careful observation. We would like to clarify that in each iteration, from the learned $\mathcal{L}_{V_i}$, we utilize Propositions 1 and 2 to identify valid edges and colliders and store them in $\mathcal{P}$. Since the edges between G and H, as well as C and F, are not validated by Propositions 1 and 2, we do not include them in $\mathcal{P}$. In other words, we only retain the information that we can confirm to be correct.

---

> > ### Comment · Reviewer_FPtp · 2025-04-02
> >
> > Thank you for addressing the feedback. After re-assessing the manuscript and evaluating the revisions, I have decided to elevate the score from 3 to 4.

---

> > > ### Author Response · Authors · 2025-04-02
> > >
> > > Thank you for your supportive feedback and for raising the score.

---

### Official Review · Reviewer_cKp3 · 2025-03-01

**Overall Recommendation:** 3

**Summary:**

The authors propose novel local characterizations that are necessary and sufficient for various types of causal relationships between two variables and bypass the need for global structure learning. Leveraging these local insights, the authors develop efficient and fully localized algorithms that accurately identify causal relationships from observational data. The authors theoretically demonstrate the soundness and completeness of the approach. Extensive experiments on benchmark networks and two real-world datasets further validate the effectiveness and efficiency of the method.

**Claims And Evidence:**

Overall, the paper is quite dense and would have been better suited for submission to a journal with a longer review period. The judgments below are based on my (possibly incorrect) understanding of its contributions.

The claims in the paper seem to be well-executed:
1. The paper clearly defines the problem of local causal structure learning with latent variables and establishes the limitations of previous approaches that assume causal sufficiency.
2. The paper provides Theorems 1-4, and proves the correctness (though I did not check them in detail).
3. The effectiveness of the proposed method is validated using synthetic benchmarks (MILDEW, ALARM, WIN95PTS, ANDES) and real-world gene expression data. However, several important baselines are missing.

**Essential References Not Discussed:**

Please see my comments above on more recent global causal discovery methods.

**Experimental Designs Or Analyses:**

See comment above.

**Methods And Evaluation Criteria:**

1. The method section is well-organized and supplemented with visual diagrams, which are very helpful.
2. However, some important baselines are missing. First, the global methods are somewhat outdated, as only PC/RFCI are considered. I encourage the authors to compare against some of more recent baselines, including [1-4]. Comparing a broader range of recent methods could provide a more comprehensive perspective on the proposed solution.
3. The benchmark datasets appear to be unconventional. If I remember correctly, bnlearn already provides a conditional probability table for forward sampling. What prevents the authors from using the default setting instead of adopting a linear Gaussian parameterization?

[1] On scoring maximal ancestral graphs with the max–min hill climbing algorithm International Journal of Approximate Reasoning 2018

[2] Iterative causal discovery in the possible presence of latent confounders and selection bias NeurIPS 2021

[3] Differentiable causal discovery under unmeasured confounding AISTATS 2021

[4] Greedy equivalence search in the presence of latent confounders UAI 2022

**Other Comments Or Suggestions:**

N/A

**Other Strengths And Weaknesses:**

Strengths:
- The paper is easy to follow given the provided visual diagram.
- I enjoyed reading the paper and the notations/terms are consistent with the literature.
- The motivation is clear and the problem studied in the paper is surely important.

Weaknesses:
- Please see my comments regarding the evaluation.
- A more thorough review of the state of the art in global causal discovery with latent confounders is missing. Most of the cited papers were published before 2020.

**Questions For Authors:**

Please consider address my concerns above during the rebuttal.

**Relation To Broader Scientific Literature:**

The paper extends an existing solution from Xie et al. and addresses an important missing piece in local causal discovery: understanding the causal relationships between variables under latent confounding without requiring global causal discovery.

**Theoretical Claims:**

The theoretical claim looks good to me.

---

> ### Author Rebuttal · Authors · 2025-04-01
>
> We sincerely appreciate the time you dedicated to reviewing our paper, as well as your insightful and encouraging comments. Below, we provide our responses to your comments.
>
>
> **Q1.** ``I encourage the authors to compare against some of more recent baselines''
>
> **A1.** Thank you for the suggestion. Within the limited time, we have added a performance comparison with M3HC [1] and ICD [2] under the same settings as in our main paper. After learning the graph using ICD and M3HC, we applied Zhang's causal identification criteria (similar to RFCI-Zhang). As shown in the table below, LocICR outperforms other methods on all evaluation metrics and significantly reduces the number of CI tests compared to ICD-Zhang, which focuses on global structure learning with latent variables. Since M3HC is a hybrid method rather than fully CI-based, we did not compare the number of CI tests.
>
> |Network|Size|Algorithm|WP↑|WR↑|WF1↑|nTest↓|
> |-|-|-|-|-|-|-|
> |**MILDEW**|1000|LocICR|**0.91**|**0.91**|**0.91**|**588.45**|
> |||M3HC-Zhang|0.83|0.82|0.80|-|
> |||ICD-Zhang|0.80|0.76|0.70|2139.98|
> ||3000|LocICR|**0.96**|**0.96**|**0.96**|**745.24**|
> |||M3HC-Zhang|0.88|0.85|0.83|-|
> |||ICD-Zhang|0.82|0.75|0.68|2857.56|
> ||5000|LocICR|**0.98**|**0.98**|**0.98**|**821.37**|
> |||M3HC-Zhang|0.86|0.85|0.84|-|
> |||ICD-Zhang|0.80|0.78|0.73|3369.91|
> |**ALARM**|1000|LocICR|**0.86**|**0.82**|**0.83**|**454.30**|
> |||M3HC-Zhang|0.85|0.79|0.80|-|
> |||ICD-Zhang|0.77|0.54|0.54|1821.39|
> ||3000|LocICR|**0.90**|**0.88**|**0.88**|**503.15**|
> |||M3HC-Zhang|0.85|0.79|0.80|-|
> |||ICD-Zhang|0.84|0.64|0.66|2231.66|
> ||5000|LocICR|**0.97**|**0.97**|**0.97**|**535.32**|
> |||M3HC-Zhang|0.88|0.82|0.83|-|
> |||ICD-Zhang|0.80|0.66|0.67|2484.25|
> |**WIN95PTS**|1000|LocICR|**0.81**|**0.77**|**0.76**|**2382.69**|
> |||M3HC-Zhang|0.77|0.70|0.69|-|
> |||ICD-Zhang|0.68|0.56|0.51|5707.26|
> ||3000|LocICR|**0.86**|**0.84**|**0.84**|**3031.96**|
> |||M3HC-Zhang|0.79|0.74|0.73|-|
> |||ICD-Zhang|0.68|0.60|0.57|7008.28|
> ||5000|LocICR|**0.92**|**0.91**|**0.91**|**3302.28**|
> |||M3HC-Zhang|0.82|0.78|0.77|-|
> |||ICD-Zhang|0.69|0.61|0.57|7639.69|
> |**ANDES**|1000|LocICR|**0.83**|**0.79**|**0.77**|**3980.06**|
> |||M3HC-Zhang|0.76|0.72|0.67|-|
> |||ICD-Zhang|0.80|0.71|0.65|76004.21|
> ||3000|LocICR|**0.85**|**0.80**|**0.78**|**4528.61**|
> |||M3HC-Zhang|0.80|0.76|0.73|-|
> |||ICD-Zhang|0.80|0.71|0.65|107370.50|
> ||5000|LocICR|**0.89**|**0.87**|**0.86**|**4931.34**|
> |||M3HC-Zhang|0.84|0.79|0.77|-|
> |||ICD-Zhang|0.81|0.72|0.67|122166.73|
> ||
>
> **Q2.** ``What prevents the authors from using the default setting instead of adopting a linear Gaussian parameterization?''
>
> **A2.** Following existing studies—specifically the setups of RFCI, LV-IDA, ITC, and IDA—we adopt a linear Gaussian parameterization, which is also used by M3HC and ICD. Additionally, we tested our method on the ALARM network with its default parameters in bnlearn, keeping other settings unchanged. The results below demonstrate that our method continues to perform well.
>
> |Size|WP↑|WR↑|WF1↑|nTest↓|
> |-|-|-|-|-|
> |1000|0.89|0.86|0.86|612.70|
> |3000|0.89|0.88|0.88|674.85|
> |5000|0.94|0.94|0.94|743.23|
> ||
>
> **Q3.** ``A more thorough review of the state of the art in global causal discovery with latent confounders is missing.''
>
> **A3.** Thank you for the valuable suggestion. In the revised version, we have included a more comprehensive review of state-of-the-art methods in global causal discovery with latent confounders, highlighting recent advancements (e.g., [1–4]).
>
> [1] On scoring maximal ancestral graphs with the max–min hill climbing algorithm International Journal of Approximate Reasoning 2018
>
> [2] Iterative causal discovery in the possible presence of latent confounders and selection bias NeurIPS 2021
>
> [3] Differentiable causal discovery under unmeasured confounding AISTATS 2021
>
> [4] Greedy equivalence search in the presence of latent confounders UAI 2022

---

### Official Review · Reviewer_sAoW · 2025-03-08

**Overall Recommendation:** 3

**Summary:**

This paper provides a local causal discovery method for inferring causal relations between a pair of variables. Specifically, given any two variables $X$ and $Y$, the proposed algorithm outputs one of the following four results: $X$ is an invariant non-ancestor of $Y$, $X$ is an explicit invariant ancestor of $Y$, $X$ is an possible ancestor of $Y$, $X$ is a implicit invariant ancestor of $Y$. The authors provide both theoretical analysis and experimental results to demonstrate the soundness and effectiveness of the proposed algorithm.


==================update after rebuttal=========================

The authors' rebuttal address most of my concerns and I have already raised my score from 2 to 3.

**Claims And Evidence:**

I think the proofs of some theoretical claims are potentially problematic, please refer to Theoretical Claims.

**Essential References Not Discussed:**

There is no related work that is essential to understanding the (context for) key contributions of the paper, but are not currently cited/discussed in the paper.

**Experimental Designs Or Analyses:**

I have checked the experimental setup and I have no major concern.

**Methods And Evaluation Criteria:**

The authors provide many theoretical claims to demonstrate the soundness of the proposed method, but I think the proofs of some theoretical claims are potentially problematic, please refer to Theoretical Claims.

**Other Comments Or Suggestions:**

N/A

**Other Strengths And Weaknesses:**

# Strengths

1. This paper investigates a novel problem: how to infer causal relations between two observed variables in the presence of hidden variables.

2. This paper provides examples for their definitions and theorems, which improves readability substantially.

# Weakness

1. Some proofs are not very clear and rigorous, please refer to Theoretical Claim.

2. This paper relies heavily on Xie et al., 2024. In fact, propositions 1, 2, 3 are all from Xie et al., 2024. This makes the theoretical contribution of this paper limited.

**Questions For Authors:**

N/A

**Relation To Broader Scientific Literature:**

The authors have discussed this in Impact Statement.

**Theoretical Claims:**

Theoretical claims in Section 5 of this paper relies heavily on theorems in Xie et al. 2024. Therefore, I have also read Xie et al. 2024 carefully, but I found some potential problems detailed in the following.

- Proposition 1 in this paper is exactly Theorem 1 in Xie et al. 2024. The proof of Theorem 1 in Xie et al. 2024 relies on Theorem 1 in Xie & Geng 2008. However, Theorem 1 in Xie & Geng 2008 presents a property of **d-separation** while Xie et al. 2024 directly replace d-separation to **m-separation** without any further clarification. I think this is a non-trivial modification. Specifically, the proof of Theorem 1 in Xie & Geng 2008 relies heavily on $An(u)$ and $An(v)$. While any variable in $An(u) \cup An(v)$ exists in the underlying DAG, some variables in $An(u) \cup An(v)$ may not exist in the underlying MAG because they may be latent variables.

- The proof of Proposition 2 is confusing. Specifically, to prove $\exists S$ s.t. $V_ 1 \perp V_ 2 | S$, the authors argue that there are three types of **active** paths between $V_ 1$ and $V_ 2$, the first and the second can be blocked by $A$ while the third can be blocked by $S_ {X, V_ 2}$. This is not valid. First, the authors should not only consider active paths, because some inactive paths my become active given $S$ if $S$ contains some colliders. Second, the authors should prove all paths are blocked by a **unique** set rather than prove different paths are provided by different sets.

- The proof of Proposition 3 is not clear (at least for me). First, according to the definition of $M_L$, it is arrived by iteratively removing leaf nodes, but when should we stop removing leaf nodes? If we don't stop, it seems that $M_L$ will be an empty graph. Second, I cannot understand why two marginal distributions equal each other ($P_{M_L}(O') = P_M(O')$) implies continuing this algorithm will not contribute to orienting the undirected edges.

Also, I have a minor question: In Zhang, 2008, PAG and maximally informative PAG are two different concepts. Does PAG in this paper actually refer to maximally informative PAG?

---

> ### Author Rebuttal · Authors · 2025-04-01
>
> We sincerely appreciate your thorough review and insightful comments. We hope the following response properly addresses your concerns.
>
> **Q1** Regarding ``...the theoretical contribution...'':
>
> **A1.** We would like to clarify that one of our paper’s main theoretical contributions is proposing necessary and sufficient local characterizations (Theorems 1–4) that account for latent variables—an aspect not found in Xie et al. (2024). To learn these local characterizations, we extend the method proposed by Xie et al., 2024 to learn $P_{MB^+(X)}$. Unlike their method, which is restricted to the target variable and its adjacent variables, we generalize it to apply to any two variables (see lines 280–285).
>
> **Q2** Regarding Proposition 1:
>
> **A2.** We would like to clarify that we have double-checked Theorem 1 in Xie et al. (2024) and confirmed its validity. A key reason is the important fact stated in Zhang (2008): given any DAG $\mathcal{G}$ over $\mathbf{V} = \mathbf{O} \cup \mathbf{L}$—where $\mathbf{O}$ denotes the set of observed variables, and $\mathbf{L}$ denotes the set of latent variables—there is a MAG over $\mathbf{O}$ alone such that for any disjoint $\mathbf{X}, \mathbf{Y}, \mathbf{Z} \subseteq \mathbf{O}$, $\mathbf{X}$ and $\mathbf{Y}$ are d-separated by $\mathbf{Z}$ in $\mathcal{G}$ if and only if they are m-separated by $\mathbf{Z}$ in the MAG. We also found similar conclusions in related works [see page 6 in Akbari et al., 2021; page 6 in Pellet & Elisseeff, 2008a].
>
> - Zhang J. Causal Reasoning with Ancestral Graphs. JMLR, 2008.
>
> **Q3** Regarding Proposition 2:
>
> **A3.** Sorry for the confusion. We would like to clarify that if $\mathbf{S}$ contains some colliders that open inactive paths, there will exist vertices belonging to $MB(X)$ similar to $A$ that can be added to $\mathbf{S}$ to block these newly activated paths. According to your suggestion, we can define $\mathbf{S}$  as the specific set $Pa^*(V_1)\cup S_{X,V_2}$. Note that when we verify how $\mathbf{S}$ blocks the three types of active paths, any collider variables introduced are also included in $\mathbf{S}$. Here, all $A_i$ belong to $Pa^*(V_1)$ and $Pa^*(V_1)\subseteq MB^+(X)$, which is observable.
>
> More specifically, the process is as follows:
>
> If there exists an active path of the form $V_1 \leftarrow A_1\dots V_2$, then $A_1\in MB(X)$ can block that path. If $A_1$ is a collider on a path $p1:$ $V_1 \dots * \rightarrow A_1 \leftarrow * \dots V_2$, due to the graph being ancestral and $V_1 \leftarrow A_1$, there must exist $V_1 \leftarrow A_2$ on $p1$. Thus $A_2\in MB(X)$ also blocks $p1$, and if $A_2$ is also a collider on a path, it falls back to the case where $A_1$ is a collider.
>
> If there exists an active path of the form $V_1 \leftrightarrow A_3 \rightarrow\dots V_2$, then $A_3\in MB(X)$ can block that path. If $A_3$ is a collider on a path $p2:$ $V_1 \dots*\rightarrow A_3 \leftarrow * \dots V_2$, due to the graph being ancestral and $A_3 \rightarrow\dots V_2$, there must exist a $A_3 \leftarrow* A_4$ on $p2$ . Thus $A_4\in MB(X)$ also blocks $p2$, and if $A_4$ is also a collider on a path, it falls back to the case where $A_3$ or $A_1$ is a collider.
>
> If there exists an active path of the form $V_1 \rightarrow\dots V_2$, then $S_{X,V_2}\subseteq MB(X)$ can block that path due to $X*\rightarrow V_1$ and $V_1 \notin S_{X,V_2}$. Thus, the edge $V_1-V_2$ is true.
>
> We will incorporate the above discussion to make the proof clearer.
>
>
> **Q4** Regarding Proposition 3:
>
> **A4.** First, if we do not stop, $M_L$ will indeed become an empty graph. We would like to clarify that lines 1055–1061 illustrate the following fact: by suitably removing leaf nodes, we can derive the local subgraph $M_L$ of interest from the global MAG $M$.
>
> Next, the reason two identical marginal distributions imply that continuing the algorithm will not further orient the undirected edges is that newly added leaf nodes do not affect the joint distribution of the existing variables in $M_L$. In other words, intuitively speaking, these leaf nodes do not introduce new directional information (e.g., V-structures) that could help further orient the undirected edges in the local structure.
>
>
> **Q5** Does PAG in this paper actually refer to maximally informative PAG?
>
> **A5.** Yes, PAG refers to the maximally informative PAG.

---

### Official Review · Reviewer_DqNE · 2025-03-14

**Overall Recommendation:** 4

**Summary:**

The paper addresses the challenge of locally identifying causal relationships between arbitrary pairs of variables in a causal graph, without assuming the absence of hidden confounders. Existing methods typically rely on access to the entire graph or impose strong assumptions about latent variables or on the pair of variables of interest, making them unsuitable for many real-world settings.
The paper provides necessary and sufficient conditions for classifying relationships as invariant ancestor, invariant non-ancestor, or possible ancestor based purely on local structure—even when latent confounders are present.
It introduces LocICR, a novel local causal discovery algorithm that determines causal relationships between a given pair of variables
The algorithm is proven to be sound and complete.
Extensive experiments on benchmark causal networks and real-world datasets demonstrate that LocICR is both effective and computationally efficient.

**Claims And Evidence:**

The authors claim that no existing method locally identifies causal relationships between arbitrary pairs of variables without assuming the absence of hidden confounders.

They provide both necessary and sufficient local characterizations for invariant ancestor, invariant non-ancestor, and possible ancestor relationships.

They introduce LocICR, a novel algorithm designed to locally infer causal relationships between variable pairs, and prove its soundness and completeness.


They demonstrate the effectiveness and efficiency of their approach on experimental data.

**Essential References Not Discussed:**

I think most relevant citation are included. I just suggested in the comment section few additional citation which can also be relevant.

**Experimental Designs Or Analyses:**

The experimental section consists of both a simulation study and a real-world data application. The simulation study is extensive and demonstrates the superior performance of the proposed method in terms of the weighted F1 score, as well as weighted precision and weighted recall. The authors also provide results on two real-world datasets, explaining these favorable outcomes with clear references.

**Methods And Evaluation Criteria:**

Yes. Rigorous experimentation has been made, 4 benchmarks have been used containing each 100 simulated datasets along with 2 real world applications.
The experimental section is clearly written, providing all the necessary details, and five other methods were used for comparison. These methods are thoroughly described.

**Other Comments Or Suggestions:**

I think the pape is very clear. But maybe consider introducing the correct definition of Markov Blanket for MAGs in the main paper (instead of deferring it to Appendix). This can make the paper even more accessible for readers unfamiliar with MAGs.


The authors are citing the seminal paper of Chekering when they first mention partial ancestral graph (PAG). I think this can be confusing since Chickering introduce a CPDAG and not a PAG (the difference between a PAG and a CPDAG substantial). At least along with this citation, cite paper truly working on PAGs such as Spirtes and Richardson, 1996; Ali et al., 2004; Zhang and Spirtes, 2005; Zhao et al., 2005.


In this sentence: "In this paper, we address the challenge of locally identifying the causal relationship between a pair of variables with- out requiring the learning of a full PAG, the enumeration of MAGs, or the assumption of causal sufficiency." You have a redundancy, if you are focusing on PAGs and MAGs then of course you are not assuming causal sufficiency.


Minor Typo in the citation: "Jonas, P., Dominik, J., and Scholkopf, B. ¨ Elements of Causal Inference. MIT Press, 2017." You replaced the first names of the first authors with their last names.


Typo: In the second line of the simulation table, “WP” should be replaced with “WR.”


Why is the performance of the proposed algorithm on MILDEW is better at small sample sizes (in WR and WP)?



Suggestion: Reference Appendix G.4 When Mentioning Benchmark Networks.

**Other Strengths And Weaknesses:**

The paper presents a compelling and relevant contribution to the field of causal inference.

It is clearly written and tackles an important problem in the domain.

The theoretical framework appears sound, and the experimental evaluation is rigorous, providing strong empirical support for the proposed approach.

By relying solely on local structure, the method remains computationally efficient and scalable, making it well-suited for large graphs where access to global knowledge is impractical.


I did not spot any major weakness other than few sentences that I found a bit confusing. More on that below.

**Questions For Authors:**

Enumerating all MAGs within a class and determining that  X is an invariant ancestor across all equivalent MAGs is equivalent to saying that X is an invariant ancestor in the PAG. No? Am I missing something? If you disagree can you please give an example? (My intuition is that PAGs represents uncertainty within a class and since in all MAGs X is an ancestor then there is no uncertainty and so this information should be visible in the PAG, assuming you are using the complete rules of FCI).
That said, I totally understand that in Malinsky and Spirtes, one way to get identification of a causal effect is via enumerating all MAGs and check if there is at least one set satisfying the generalized back-door criterion in each MAG. However finding a set satisfying the back-door criterion is not the same as finding ancestors. The latter is more complicated because the information it requires is not always visible by looking at the PAG. Maybe I am missing something so please correct me if you think I am wrong. I have scored the paper assuming that I am missing something related to this, I will update my score based on the rebuttal.


Since you are interested in Local causal discovery, is it possible to reduce the faithfulness assumption into a "local" faithfulness?

**Relation To Broader Scientific Literature:**

The authors are claiming that they are proposing the first method that locally identifies causal relationships between arbitrary pairs of variables without assuming the absence of hidden confounders (To the best of my knowledge this is true). Which means this paper is making causal inference more applicable in real world settings. Especially when the true causal graph is unknown and where discovering the true entire graph can be complicated for many reasons ...

**Theoretical Claims:**

I checked some proofs and they seem to be sound.

---

> ### Author Rebuttal · Authors · 2025-04-01
>
> We sincerely appreciate your constructive and thoughtful feedback, as well as your recognition of the importance, clarity, and empirical rigor of our work in the field of causal inference. We hope the following responses properly address your concerns.
>
>
> **Q1** ``introducing the correct definition of Markov Blanket for MAGs in the main paper''
>
> **A1.** Following your suggestion, we have moved the definition of the Markov Blanket for MAGs from the Appendix into the main text.
>
> **Q2** ``citing the seminal paper of Chekering when they first mention PAG''
>
> **A2.** Thank you for noting this. It was indeed a citation error, which we have now corrected in the revised version. We have properly cited works specifically addressing PAGs, including [Spirtes and Richardson, 1996], [Ali et al., 2004], [Zhang and Spirtes, 2005], and [Zhao et al., 2005].
>
> **Q3** ``... without requiring the learning of a full PAG, the enumeration of MAGs, or the assumption of causal sufficiency.'' Having a redundancy.
>
> **A3.** Thank you for your thoughts. We have removed this redundancy in the revised version. Our initial aim was to emphasize that the proposed approach does not rely on the assumption of causal sufficiency.
>
> **Q4** ``Why is the performance of the proposed algorithm on MILDEW is better at small sample sizes (in WR and WP)?''
>
> **A4.** Because our algorithm reduces the number of conditional independence (CI) tests through localization, it helps avoid bias introduced by excessive testing. Consequently, even with smaller sample sizes, our method outperforms other global approaches. Notably, existing local methods often perform worse because they require the assumption of causal sufficiency, while latent variables may be present in the system.
>
> **Q5** ``Enumerating all MAGs within a class and determining that X is an invariant ancestor across all equivalent MAGs is equivalent to saying that X is an invariant ancestor in the PAG?''
>
> **A5.** You are correct! Stating that X is an invariant ancestor in  all equivalent MAGs is equivalent to saying that X is an invariant ancestor in the PAG.
>
> **Q6** ``My intuition is that PAGs represents uncertainty within a class and since in all MAGs X is an ancestor then there is no uncertainty and so this information should be visible in the PAG, assuming you are using the complete rules of FCI. ... However finding a set satisfying the back-door criterion is not the same as finding ancestors. ''
>
> **A6.** First, we would like to clarify that although $X$ may be an ancestor in all equivalent MAGs, this fact might not always be explicitly visible(i.e., by checking the directed path) in the PAG. This discrepancy arises because PAGs represent uncertainty within a class of MAGs and do not explicitly encode all ancestral relationships. For instance, as illustrated in Fig. 6, there is no directed path from $A$ to $E$ in the PAG shown in Fig. 6(b), yet $A$ is an ancestor of $E$ in all equivalent MAGs (Fig. 6(c)-(g)).
>
> Furthermore, LV-IDA enumerates all (local) MAGs, identifies adjustment sets that satisfy the generalized back-door criterion, and estimates the possible causal effects accordingly. If all such causal effects are non-zero, the method concludes an invariant ancestor (Fang et al., 2022). For instance, in Fig. 6, LV-IDA calculates the causal effect from $A$ to $E$ for every equivalent MAG (Fig. 6(c)-(g)) as non-zero, thereby concluding that $A$ is an invariant ancestor of $E$.
>
>
> **Q7** ``is it possible to reduce the faithfulness assumption into a "local" faithfulness?''
>
> **A7.** Thank you for your thoughtful consideration. The local learning—such as our approach—can reduce unnecessary CI tests, which may help mitigate minor violations of the faithfulness assumption [Isozaki, 2014]. We will discuss this point in our paper and plan to explore it more thoroughly in future work.
>
> Isozaki T. A robust causal discovery algorithm against faithfulness violation[J]. Information and Media Technologies, 2014, 9(1): 121-131.
>
> > Thank you again for your careful reading. We have corrected the typos in the citations and simulation table in the revised version.

---

### Decision · Program_Chairs · 2025-05-01

**Decision:**

Accept (spotlight poster)

**Comment:**

This paper introduces LocICR, a local method for identifying causal relationships between arbitrary pairs of variables in the presence of latent confounders. Unlike prior approaches that rely on learning a global causal structure such as a PAG or MAG, the proposed method operates entirely locally, leveraging new necessary and sufficient conditions for distinguishing invariant ancestor, non-ancestor, and possible ancestor relationships. The authors demonstrate soundness and completeness of their characterizations and provide an efficient algorithm that avoids the computational overhead of global methods.

The paper is well written and makes a strong theoretical and practical contribution. Several reviewers praised the clarity of the local characterizations and the extensive experimental evaluation, which covers both simulated benchmarks and real-world datasets. The authors also responded thoroughly during the rebuttal phase, addressing theoretical concerns and clarifying their distinctions from prior work. In particular, they provided detailed explanations of their proof dependencies, clarified differences with methods like IDA and LV-IDA, and included additional baseline comparisons with more recent global discovery methods (e.g., M3HC, ICD), strengthening the empirical case.

While one reviewer questioned the novelty of some of the technical components due to their relation to Xie et al. (2024), the authors convincingly showed how their generalization to arbitrary variable pairs and local characterizations under latent confounding go beyond previous work. Other concerns about unclear proofs and experimental setup were addressed to the satisfaction of the reviewers, with several increasing their scores post-rebuttal.

Overall, this is a rigorous and impactful paper that advances the state of local causal discovery, particularly in the challenging setting where latent variables are present. It offers a practical and theoretically grounded alternative to global structure learning and is well-suited for applications where local causal queries are more relevant than recovering the full graph. Therefore, I recommend acceptance.